# Vps34 PI 3-kinase inactivation enhances insulin sensitivity through reprogramming of mitochondrial metabolism

Benoit Bilanges[1], Samira Alliouachene[1], Wayne Pearce[1], Daniele Morelli[1], Gyorgy Szabadkai [2,3], Yuen-Li Chung[4], Gaëtan Chicanne[5], Colin Valet[5], Julia M. Hill[2], Peter J. Voshol[6], Lucy Collinson[7], Christopher Peddie[7], Khaled Ali[1], Essam Ghazaly [8], Vinothini Rajeeve[8], Georgios Trichas[9], Shankar Srinivas[9], Claire Chaussade[1], Rachel S. Salamon[10], Jonathan M. Backer[10], Cheryl L. Scudamore[11], Maria A. Whitehead[1], Erin P. Keaney[12], Leon O. Murphy[12], Robert K. Semple[13], Bernard Payrastre[5], Sharon A. Tooze[7] & Bart Vanhaesebroeck[1]

Vps34 PI3K is thought to be the main producer of phosphatidylinositol-3-monophosphate, a lipid that controls intracellular vesicular trafficking. The organismal impact of systemic inhibition of Vps34 kinase activity is not completely understood. Here we show that heterozygous Vps34 kinase-dead mice are healthy and display a robustly enhanced insulin sensitivity and glucose tolerance, phenotypes mimicked by a selective Vps34 inhibitor in wild-type mice. The underlying mechanism of insulin sensitization is multifactorial and not through the canonical insulin/Akt pathway. Vps34 inhibition alters cellular energy metabolism, activating the AMPK pathway in liver and muscle. In liver, Vps34 inactivation mildly dampens autophagy, limiting substrate availability for mitochondrial respiration and reducing gluconeogenesis. In muscle, Vps34 inactivation triggers a metabolic switch from oxidative phosphorylation towards glycolysis and enhanced glucose uptake. Our study identifies Vps34 as a new drug target for insulin resistance in Type-2 diabetes, in which the unmet therapeutic need remains substantial.

[1] UCL Cancer Institute, University College London, 72 Huntley Street, London WC1E 6DD, UK. [2] Department of Cell and Developmental Biology, Consortium for Mitochondrial Research, University College London, Gower Street, London WC1E 6BT, UK. [3] Department of Biomedical Sciences, University of Padua, Padua, 58/B via Ugo, Bassi 35121, Italy. [4] Cancer Research UK Cancer Imaging Centre, Division of Radiotherapy and Imaging, The Institute of Cancer Research London, 123 Old Brompton Road, London SW7 3RP, UK. [5] Inserm/UPS UMR 1048, Institut des Maladies Métaboliques et Cardiovasculaires, 1 Avenue Jean Poulhès BP 84225, 31432 Toulouse Cedex 4, France. [6] Metabolic Research Laboratories, MRC Metabolic Diseases Unit, Wellcome Trust-MRC Institute of Metabolic Science, Level 4, Box 289, Addenbrooke's Hospital, Cambridge CB2 0QQ, UK. [7] The Francis Crick Institute, Lincoln's Inn Fields Laboratories, 44 Lincoln's Inn Fields, London WC2A 3LY, UK. [8] Centre of Haemato-Oncology, Barts Cancer Institute, Queen Mary University of London, London EC1M 6BQ, UK. [9] Department of Physiology Anatomy and Genetics, University of Oxford, Oxford OX1 3PT, UK. [10] Department of Molecular Pharmacology, Albert Einstein College of Medicine, Bronx 10461 NY, USA. [11] Mary Lyon Centre, MRC Harwell, Harwell Science and Innovation Campus, Harwell OX11 0RD, UK. [12] Novartis Institutes for BioMedical Research, 181 Massachusetts Avenue, Cambridge, MA 02139, USA. [13] Institute of Metabolic Science, University of Cambridge, Addenbrooke's Hospital, Cambridge CB2 0QQ, UK. Correspondence and requests for materials should be addressed to B.B. (email: b.bilanges@ucl.ac.uk) or to B.V. (email: bart.vanh@ucl.ac.uk)

The three classes of PI3K phosphorylate phosphoinositides, a group of lipids that modulate multiple cellular processes[1]. In contrast to the class I PI3K isoforms, which have been implicated in signaling and disease[2], the physiological roles of the class II/III PI3K family members remain enigmatic. The class III PI3K, Vps34, is the primordial isoform of PI3K that is conserved from yeast to human. Vps34 converts the phosphatidylinositol (PI) membrane lipid to PI3P, which coordinates the localization and function of effector proteins containing PI3P-binding domains such as FYVE, PX, or the FRRG domain found in PROPPINS, thereby controlling PI3P-mediated intracellular vesicular trafficking[3]. This includes (1) the earliest steps in the autophagy process where PI3P generation is a key event in autophagosome biogenesis, as well as later steps in autophagosome maturation, (2) endosomal maturation, and (3) phagocytosis[3]. Vps34 is present in multiple protein complexes. Whereas complex I functions in autophagy and contains Vps34, Vps15, Beclin-1, and Atg14, complex II takes part in endocytic sorting and contains the same components as complex I, except that Atg14 is replaced by UVRAG[3].

Homozygous Vps34 gene knockout (KO) in mice reveals that this PI3K is indispensable for embryogenesis, organ function and cell survival[4–14]. However, in addition to its catalytic activity, Vps34 also has a scaffolding function in the assembly of the different Vps34 complexes. This has confounded the interpretation of the phenotypes observed in Vps34 KO mice, as the expression of most of the proteins that form the distinct Vps34 complexes are severely reduced upon loss of Vps34 expression. In this study, we set out to uncover the role of the catalytic activity of Vps34 in organismal metabolism, with a view to genetically model the impact of pharmacological inactivation of this kinase.

Gene KO approaches, either ubiquitous or tissue-specific, are unlikely to mirror the effects of systemic administration of a pharmaceutical inhibitor because a drug almost invariably inhibits the kinase without affecting its expression. We thus created mice in which the kinase activity of Vps34 was disabled in the germline, by the introduction of a kinase-inactivating point mutation in the DFG motif of the kinase domain, as we previously reported for other isoforms of PI3K[15–20]. To further model the pharmacological effect of kinase inhibition, which is most often incomplete in vivo, we focused on the study of mice with heterozygous inactivation of Vps34.

## Results

**Generation of Vps34 kinase-dead knockin mice.** Vps34 occurs in distinct multi-protein complexes that exert specific biological functions[3]. Abrogation of Vps34 protein expression, as observed in Vps34 KO mice[5], reduces the expression of multiple components of these Vps34 complexes, resulting in complex biological "knock-on" phenotypes. Vps34 gene deletion studies thus assess both the scaffolding and kinase-dependent functions of Vps34.

To specifically uncover the role of the catalytic activity of Vps34, we introduced a germline kinase-inactivating knockin (KI) mutation into the Vps34-encoding *Pik3c3* gene, resulting in conversion of the critical DFG sequence in the ATP-binding site of the Vps34 protein to AFG (called hereafter D761A), giving rise to the Vps34^D761A protein (Supplementary Fig. 1a). This is expected to constitutively inactivate the kinase activity of Vps34, without disrupting Vps34 protein expression. This strategy, which we previously applied to other PI3K isoforms[15–20], better mimics the impact of a systemically administered small molecule ATP-competitive kinase inhibitor than a gene KO strategy.

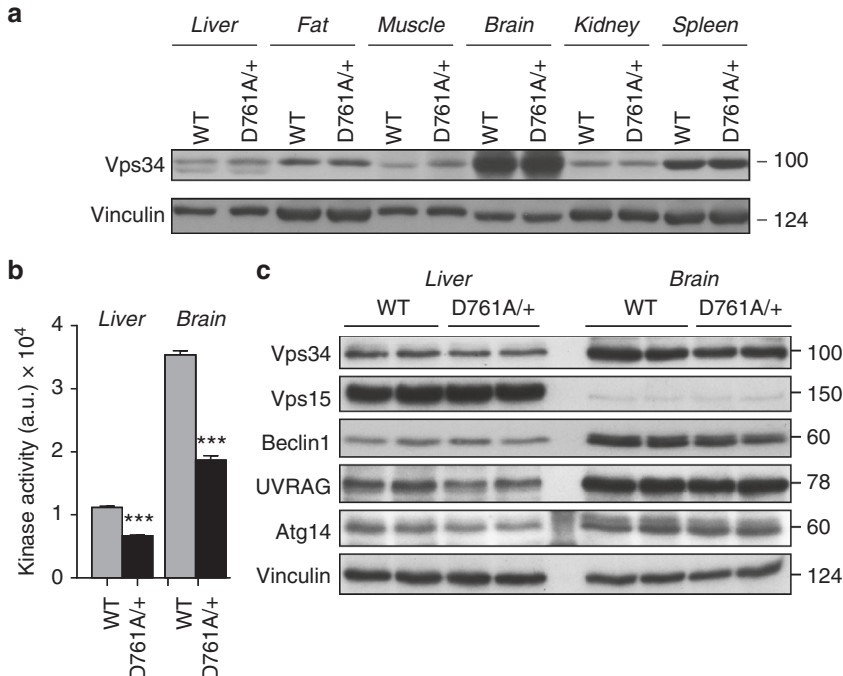

**Fig. 1** Characterization of Vps34^D761A/+ mice. **a** Expression of the Vps34 protein in WT and Vps34^D761A/+ mouse tissues. 100 μg of protein was loaded per lane. Representative data from three independent experiments. **b** Lipid kinase activity associated with Vps34 in WT and Vps34^D761A/+ mouse tissues. Tissue and cell homogenates were immunoprecipitated with an anti-Vps34 antibody, and an in vitro lipid kinase assay was performed using PI as substrate. Data represent mean ± SEM (non-parametric Mann–Whitney *t*-test) *$p < 0.05$, **$p \leq 0.01$, ***$p \leq 0.001$. 4 mice/genotype. **c** Expression levels of Vps34 and its binding partners in liver and brain tissues of 12-week-old mice. Tissue/cell lysates were immunoblotted using the indicated antibodies. 60 μg of protein was loaded per lane. Composite images are derived from experiments whereby equal amounts of the same cell/tissue extract were loaded on separate gels and developed with the same indicated antibodies. This was done to quantify proteins with a similar molecular weight

Consistent with previous studies[21], we found Vps34 to be ubiquitously expressed in adult mouse tissues, with the highest expression in the brain (Fig. 1a). In heterozygous Vps34[D761A/+] mice, which were viable and fertile, Vps34 lipid kinase activity was reduced by ~50% (Fig. 1b), without changes in the expression levels of Vps34 and that of its binding partners, in primary tissues (Fig. 1a, c) and in isolated Vps34[D761A/+] cells, such as mouse embryonic fibroblasts (MEFs) and primary hepatocytes (Supplementary Fig. 1b, c).

Homozygous Vps34[D761A/D761A] mice died between embryonic day (E) 6.5 and 8.5 (Supplementary Table 1; Supplementary Fig. 2a), confirming a critical role of Vps34 in embryogenesis[13]. Vps34[D761A/D761A] blastocysts appeared morphologically normal at E3.5 + 1, but showed defective in vitro outgrowth of both inner cell mass and trophoblast cells at later time points (Supplementary Fig. 2b). At present, the underlying molecular mechanism of this early embryonic lethality is unknown.

**Metabolic improvement upon Vps34 inactivation in vivo.** Heterozygous Vps34[D761A/+] mice were born at expected Mendelian ratios (Supplementary Table 1), and had no behavioral defects or abnormalities in overall histopathology in 37 tissues investigated at the age of 3 month (Supplementary Table 2). No apparent behavioral abnormalities were detected in aged Vps34[D761A/+] mice as compared with the littermate WT controls, tested up to 54 months.

Previous cell-based studies[22, 23] suggested that Vps34 positively controls insulin-stimulated activation of the protein kinase S6K1, which is activated in response to nutritional status and hormonal stimulation to regulate organismal glucose metabolism[24]. We thus set out to investigate whether glucose metabolism was affected in Vps34[D761A/+] mice. Under normal chow-fed diet (NCD), WT, and Vps34[D761A/+] mice showed no differences in body weight (Supplementary Fig. 3a), food intake (Supplementary Fig. 3b), fat to lean mass ratio (Supplementary Fig. 3c), and blood glucose levels (Fig. 2a). However, partial Vps34 inactivation reduced fasted glycaemia (~16% decrease) (Fig. 2a), with plasma insulin levels remaining unchanged in both fed and fasted mice (Fig. 2b). These data suggested that glucose metabolism might be affected. We therefore performed insulin and glucose tolerance tests (ITTs and GTTs). Surprisingly, Vps34[D761A/+] mice showed improved glucose clearance compared with the WT mice (Fig. 2c) and a higher insulin sensitivity (Fig. 2d).

**Vps34 inactivation protects against HFD-induced steatosis.** To investigate the role of Vps34 kinase activity in a more pathophysiological context, we subjected WT and Vps34[D761A/+] mice to 16 weeks of high-fat diet (HFD; 45% fat). Under these conditions, WT and Vps34[D761A/+] mice showed a similar gain in body weight, although Vps34[D761A/+] mice had a tendency to be leaner (difference in the area under the curve between WT and Vps34[D761A/+] mice: $p < 0.0577$; non-parametric Mann–Whitney $t$-test) (Supplementary Fig. 3d). Compared to WT mice, Vps34[D761A/+] mice showed no differences in fed glycaemia but, as for NCD conditions, displayed a significant reduction in fasting glycaemia (Fig. 3a), with unaltered insulin levels in both fed and fasting states (Fig. 3b). HFD-fed Vps34[D761A/+] mice had improved glucose tolerance (Fig. 3c) and better insulin sensitivity (Fig. 3d) than HFD-fed WT mice.

Compared to control mice, Vps34[D761A/+] mice also showed a marked reduction in HFD-induced hepatic steatosis, a condition strongly associated with insulin resistance and Type-2 diabetes[25], as well as a reduction of neutral lipids (as assessed by Oil Red O staining) (Fig. 3e, f). This correlated with a significant decrease in the weight of the liver (Fig. 3g), but not that of other tissues

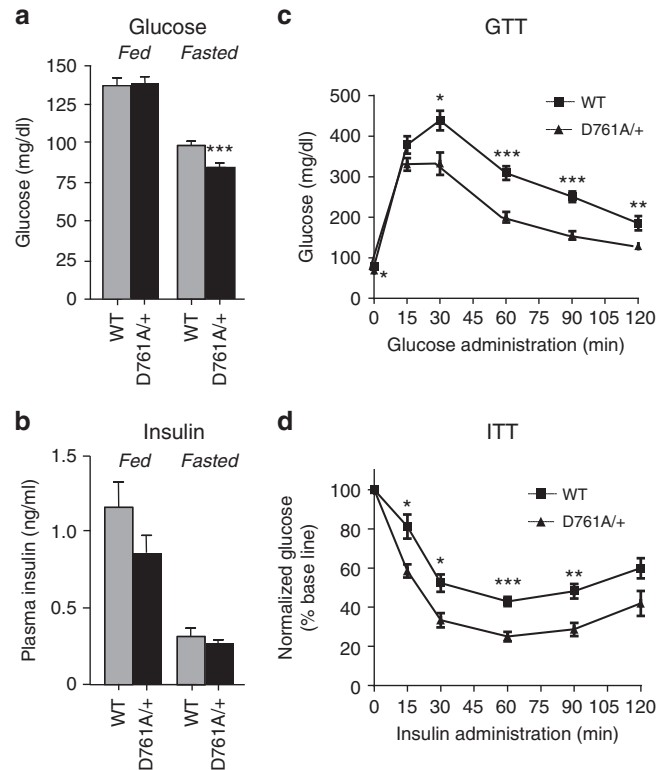

**Fig. 2** Impact of Vps34 inactivation on glucose tolerance, insulin sensitivity in lean mice. **a** Serum glucose levels under randomly fed and fasted conditions. Data represent mean ± SEM (non-parametric Mann–Whitney $t$-test). **b** Serum insulin levels under randomly fed and fasted conditions. Data represent mean ± SEM (non-parametric Mann–Whitney $t$-test). **c** GTT after intraperitoneal injection of 2 g/kg of glucose in mice after overnight starvation. Data represent mean ± SEM (non-parametric Mann–Whitney $t$-test). **d** ITT after intraperitoneal injection of 0.75 U/kg of insulin in mice after overnight starvation. Data represent mean ± SEM (non-parametric Mann–Whitney $t$-test). For all experiments shown, ≥10 mice/genotype were used. *$p < 0.05$, **$p \leq 0.01$, ***$p \leq 0.001$

(Supplementary Fig. 3e), as well as decreased levels of triglycerides in the liver and plasma (Fig. 3h, i) and of plasma cholesterol (Supplementary Fig. 3f) in HFD-fed Vps34[D761A/+] mice. The levels of serum adiponectin, an adipokine that reduces hepatic and serum triglyceride levels and protects from hepatic steatosis[26], were significantly increased in HFD-fed Vps34[D761A/+] mice (Supplementary Fig. 3g), whereas the levels of leptin, another adipokine that controls energy balance and body weight, were unchanged (Supplementary Fig. 3h). Taken together, these data reveal that Vps34 kinase activity negatively regulates insulin sensitivity and glucose metabolism in vivo.

**Vps34 inactivation reduces hepatic glucose production.** To identify the tissues responsible for the glucose-lowering effect observed in Vps34[D761A/+] mice, we performed hyperinsulinaemic-euglycaemic clamp and in vivo glucose uptake experiments. Hepatic glucose production was lower in Vps34[D761A/+] mice under both normal chow and HFD conditions (Fig. 4a).

**Vps34 inactivation increases glucose uptake in muscle.** In addition, glucose uptake was increased in skeletal muscle of Vps34[D761A/+] mice, reaching statistical significance in HFD-fed mice (Fig. 4b), with a similar tendency seen in brown (but not white) adipose tissue under both diets (Supplementary Fig. 3i, j).

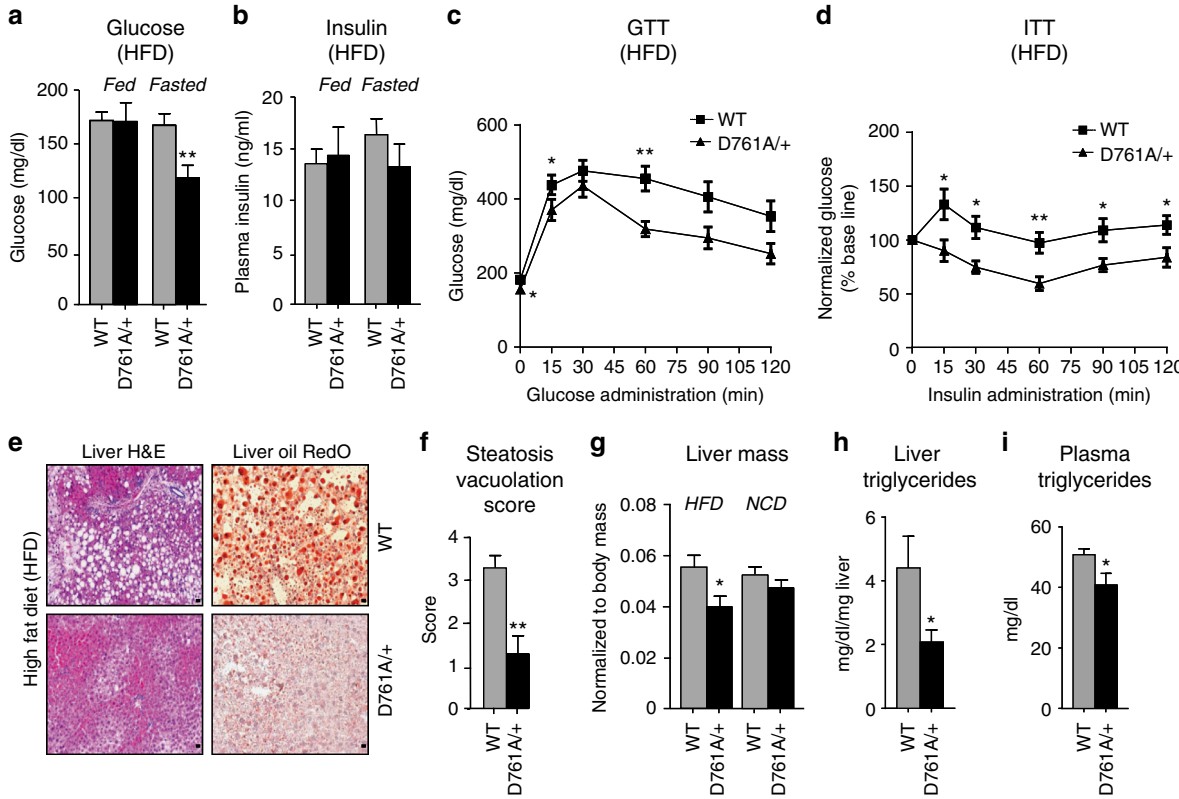

**Fig. 3** Impact of Vps34 inactivation on glucose tolerance, insulin sensitivity in HFD-fed mice. **a** Serum glucose levels in randomly fed and fasted mice. Data represent mean ± SEM (non-parametric Mann–Whitney *t*-test). **b** Serum insulin levels in randomly fed and fasted mice. Data represent mean ± SEM (non-parametric Mann–Whitney *t*-test). **c** GTT after intraperitoneal injection of 2 g/kg of glucose in mice in overnight- starved mice. Data represent mean ± SEM (non-parametric Mann–Whitney *t*-test). **d** ITT after intraperitoneal injection of 0.75 U/kg of insulin in overnight-starved mice. Data represent mean ± SEM (non-parametric Mann–Whitney *t*-test). **e** Hematoxylin and eosin (H&E) staining and Oil Red O staining of liver sections of mice after 16 weeks of HFD. Representative image derived from 6 livers/genotype of mice. Scale bar, 10 μm. **f** Quantification of vacuolation of 6 livers/genotype of mice after 16 weeks of HFD. a.u., arbitrary units. Data represent mean ± SEM (non-parametric Mann–Whitney *t*-test). **g** Comparison of liver weight of mice fed a NCD or HFD. Data represent mean ± SEM (non-parametric Mann–Whitney *t*-test). Quantification of 5–9 livers/genotype. **h** Triglyceride levels in liver of mice challenged with 45% HFD for 16 weeks. Data represent mean ± SEM (non-parametric Mann–Whitney *t*-test). **i** Plasma triglyceride levels of mice challenged with 45% HFD for 16 weeks. Data represent mean ± SEM (non-parametric Mann–Whitney *t*-test). For all experiments shown, ≥10 mice/ genotype were used. $*p < 0.05$, $**p \le 0.01$, $***p \le 0.001$

Taken together, these findings show that heterozygous inactivation of Vps34 enhances glucose tolerance and insulin sensitivity by reducing glucose production in the liver and stimulating glucose uptake in muscle.

**Vps34 inactivation mildly reduces autophagy in the liver**. We next sought to uncover the cellular processes that Vps34 activity regulates to control organismal glucose metabolism. Vps34 is considered to be the main producer of cellular PI3P to regulate autophagy and contributes to the PI3P pool for endocytic traffic[3]. Surprisingly, the total cellular levels of PI3P, as measured by mass assay[27], were not significantly altered in tissues of NCD-fed or starved Vps34[D761A/+] mice (Fig. 5a, b) and in primary Vps34[D761A/+] hepatocytes cultured in absence of insulin (hereafter called insulin-starved condition) (Fig. 5c).

Confocal microscopy using the GST-2XFYVE[HRS] PI3P-binding probe[28], a more sensitive approach to quantitatively and qualitatively monitor the levels and distribution of PI3P than the PI3P mass assay, revealed an impact of heterozygous Vps34 inactivation on subcellular PI3P pools. The FYVE puncta were reduced (~26%) in number (Fig. 5d) and mildly increased (~10%) in size (Supplementary Fig. 4a) in insulin-starved Vps34[D761A/+] hepatocytes compared to WT cells. A similar decrease (~21%)

was found in the number of puncta of the endosomal PI3P-binding effector EEA1 (Fig. 5d), without an effect on EEA1 puncta size (Supplementary Fig. 4a).

Importantly, analysis of autophagy in Vps34[D761A/+] hepatocytes revealed a reduction in the puncta number of WIPI-2, a PI3P-binding protein involved in the initiation of autophagy[29], both under non-starved and amino acid-starved conditions (30 and 40% reduction, respectively; Fig. 5e). In agreement with this, we also observed mild but statistically significant reductions in LC3 lipidation and p62 levels under non-starved and amino acid-starved conditions in Vps34[D761A/+] primary hepatocytes (Fig. 5f; Supplementary Fig. 4b) and reductions in p62 levels in vivo in liver tissue of Vps34[D761A/+] mice (Fig. 5g, Supplementary Fig. 4c). Despite an effect on LC3 lipidation and p62 levels, we did not observe a difference in the number of LC3 puncta between WT and Vps34[D761A/+] hepatocytes under non-starved and amino acid-starved conditions (Supplementary Fig. 4d). These observations show that inactivation of 50% of Vps34 kinase activity results in a reduction in endosomal (EEA1-associated) and autophagic (WIPI-2-associated) pools of PI3P-containing vesicles, with partially dampening overall autophagy. Qualitative and quantitative electron microscopy analysis did not reveal any robust quantitative differences between WT and Vps34[D761A/+] primary hepatocytes under starved conditions, confirming our

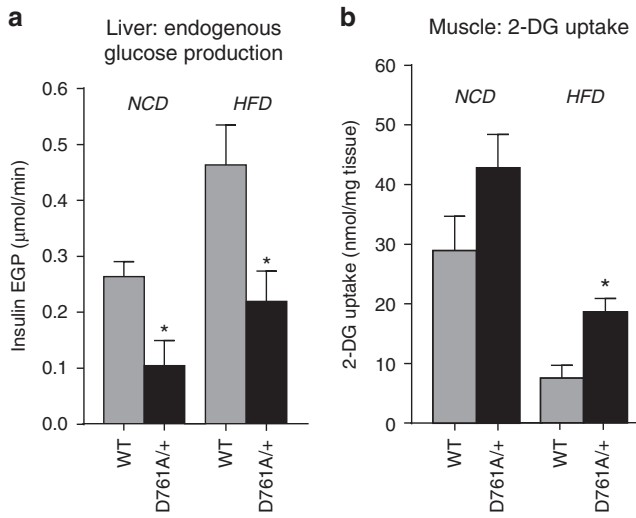

**Fig. 4** Impact of Vps34 inactivation on hepatic glucose production and glucose uptake in muscle in vivo. **a** Endogenous glucose production (EGP) in liver in hyperinsulinemic-euglycemic clamp experiments in WT ($n = 5$) and Vps34$^{D761A/+}$ mice ($n = 6$) under NCD, and in WT ($n = 7$) and Vps34$^{D761A/+}$ mice ($n = 6$) under HFD. Glucose kinetics are shown in Supplementary Table 3. Data represent mean ± SEM (non-parametric Mann–Whitney t-test). **b** 2-Deoxy-D-glucose (2-DG) uptake in muscle in hyperinsulinemic-euglycemic clamp experiments in WT ($n = 3$) and Vps34$^{D761A/+}$ mice ($n = 6$) under NCD, and in WT ($n = 5$) and Vps34$^{D761A/+}$ mice ($n = 7$) under HFD. Data represent mean ± SEM (non-parametric Mann–Whitney t-test). *$p < 0.05$, **$p \leq 0.01$, ***$p \leq 0.001$

data (Fig. 5e–g; Supplementary Figs. 4b–d and 5) that 50% inactivation of Vps34 kinase activity does not fully abolish the formation of autophagosomes and autophagolysosomes under starvation (Supplementary Fig. 5).

Taken together, these data confirm that Vps34 kinase activity drives and coordinates endosomal and autophagy trafficking pathways.

**Reduced mitochondrial respiration upon vps34 inactivation.** We next investigated the impact of reduced Vps34 activity on cellular metabolism. As systemic glucose and lipid metabolism converge on mitochondria to generate the majority of cellular ATP, we focused on the impact of Vps34 inactivation on this organelle. Compared to WT cells, Vps34$^{D761A/+}$ hepatocytes had increased mitochondrial content, as assessed by visualization using the Tom20 mitochondrial marker (Fig. 6a), quantitative FACS analysis using mitotracker FM dye, which stains mitochondria regardless of mitochondrial membrane potential (Supplementary Fig. 6a) and the levels of mitochondrial proteins involved in oxidative phosphorylation (Supplementary Fig. 6b).

To assess mitochondrial function, we first measured mitochondrial respiration in intact cells (see Supplementary Fig. 6c for schematic). The oxygen consumption rate (OCR) was significantly reduced in Vps34$^{D761A/+}$ hepatocytes and myotubes, compared to WT cells (Fig. 6b, c). This was also observed in the Hepa1.6 hepatoma cell line upon treatment with Vps34-IN1, a highly selective inhibitor of Vps34[30, 31,] (Supplementary Fig. 6d). In addition to reduced mitochondrial respiration, Vps34$^{D761A/+}$ myotubes showed a clear increase in glycolysis compared to WT cells (Fig. 6d). Correlating with these findings, we found that total in vivo ATP levels were significantly reduced in the liver (Fig. 6e) but unchanged, or even mildly increased, in skeletal muscle of Vps34$^{D761A/+}$ mice (Fig. 6e). This could be due to the fact that muscle, unlike the liver, can produce ATP from

sources other than mitochondria (glycolysis and possibly phosphocreatinine) to cope with its high energy demands, potentially masking the reduction in mitochondrial ATP production observed in Vps34$^{D761A/+}$ myotubes in vitro (Fig. 6d). To circumvent compensatory mechanisms that could be induced by long-term inactivation of Vps34, we next tested whether acute pharmacological inhibition of Vps34 altered ATP levels in the murine C2C12 myoblast and Hepa1.6 hepatoma cell lines. As shown in Supplementary Fig. 6e, inhibition of Vps34 markedly reduced the ATP levels in these cells, further indicating that Vps34 activity can control ATP levels in cells of muscle origin.

To assess whether the impaired mitochondrial respiration observed in Vps34$^{D761A/+}$ hepatocytes was due to deficient electron transport, we measured oxygen consumption in freshly isolated mitochondria from the liver. As shown in Fig. 7, ATP synthase-coupled respiration (state 3), driven either by respiratory complex I (glutamate/pyruvate/malate) or complex II (succinate + rotenone), did not differ between isolated hepatic mitochondria from WT and Vps34$^{D761A/+}$ mice. This suggests that in isolated mitochondria (as compared to mitochondria in intact cells), mitochondrial oxygen consumption and substrate-dependent respiration were not compromised by Vps34 inactivation. This suggests that intrinsic mitochondrial function is not impaired, but that intracellular factor(s), or lack of these due to Vps34 inactivation, alter mitochondrial respiration in intact cells.

Altogether, these data indicate that inhibition of Vps34 switches off mitochondrial respiration, leading to the activation of ATP-producing pathways such as glycolysis, which depend on glucose uptake.

**Vps34 blockade reduces substrate availability for respiration.** On the basis of the observations above, we hypothesized that Vps34 inactivation might alter the intracellular availability of substrates important for the TCA cycle to function properly. Amino acids are the major products released by starvation-induced autophagy and can be catabolized in mitochondria for ATP production via the TCA cycle or, alternatively, be used to synthesize glucose. NMR studies on liver extracts of WT and Vps34$^{D761A/+}$ mice subjected to 20 h starvation revealed a significant reduction in the levels of specific amino acids (threonine and alanine) as well as reductions in the levels of glucose and lactate upon Vps34 inactivation (Fig. 8a, b; Supplementary Fig. 6f). The reduced levels of threonine and alanine, anaplerotic metabolites that replenish the TCA cycle, are likely to explain the reduced mitochondrial respiration observed in Vps34$^{D761A/+}$ cells. Importantly, threonine. alanine and lactate are also important glucogenic metabolites, and their reduced levels are likely responsible for the attenuated hepatic gluconeogenesis in Vps34$^{D761A/+}$ mice upon starvation.

**Improved AMPK signaling upon Vps34 inhibition.** We next investigated the impact of Vps34 inactivation on signaling pathways involved in insulin (PI3K/Akt/mTORC1) and energy (AMPK) sensing.

Both basal and insulin-stimulated activation of Akt (as assessed by phosphorylation of Akt on S473 and T308) and its downstream targets such as GSK3α/β (on S21 and S29), AS160 (on T642), and PRAS40 (on T246)) and mTORC1 (assessed by phosphorylation of S6K (on T389) and S6 (on S240 and 244)), were unaffected in Vps34$^{D761A/+}$ myotubes (Fig. 9a; Supplementary Fig. 7a) and hepatocytes (Supplementary Fig. 7b–d), as well as in muscle, liver and white adipose tissue (WAT) in vivo (Supplementary Fig. 7e–g). Similar observations were made upon treating WT hepatocytes with the Vps34-IN1 inhibitor (Supplementary Fig. 7b). Taken together, these data demonstrate that the

insulin sensitization observed in Vps34$^{D761A/+}$ mice is independent of insulin-mediated Akt/mTORC1 signaling. This contrasts with previous cell-based studies using Vps34 RNAi that concluded that Vps34 positively controls insulin-stimulated activation of S6K1[22, 23].

In contrast to the unaltered Akt/mTORC1 signaling, activation of AMPK (as measured by the activating phosphorylation of T172

in the AMPK activation loop) was enhanced in Vps34$^{D761A/+}$ myotubes (Fig. 9a; Supplementary Fig. 7a) and hepatocytes (Fig. 9b), compared to WT cells. In agreement with this, a significant enhancement in the phosphorylation of AMPK substrates, such as Acetyl-CoA carboxylase (ACC; on S79) and TBC1D1 (on T660 and S237) was observed in the basal state, in Vps34$^{D761A/+}$ myotubes and muscle (Fig. 9a; Supplementary

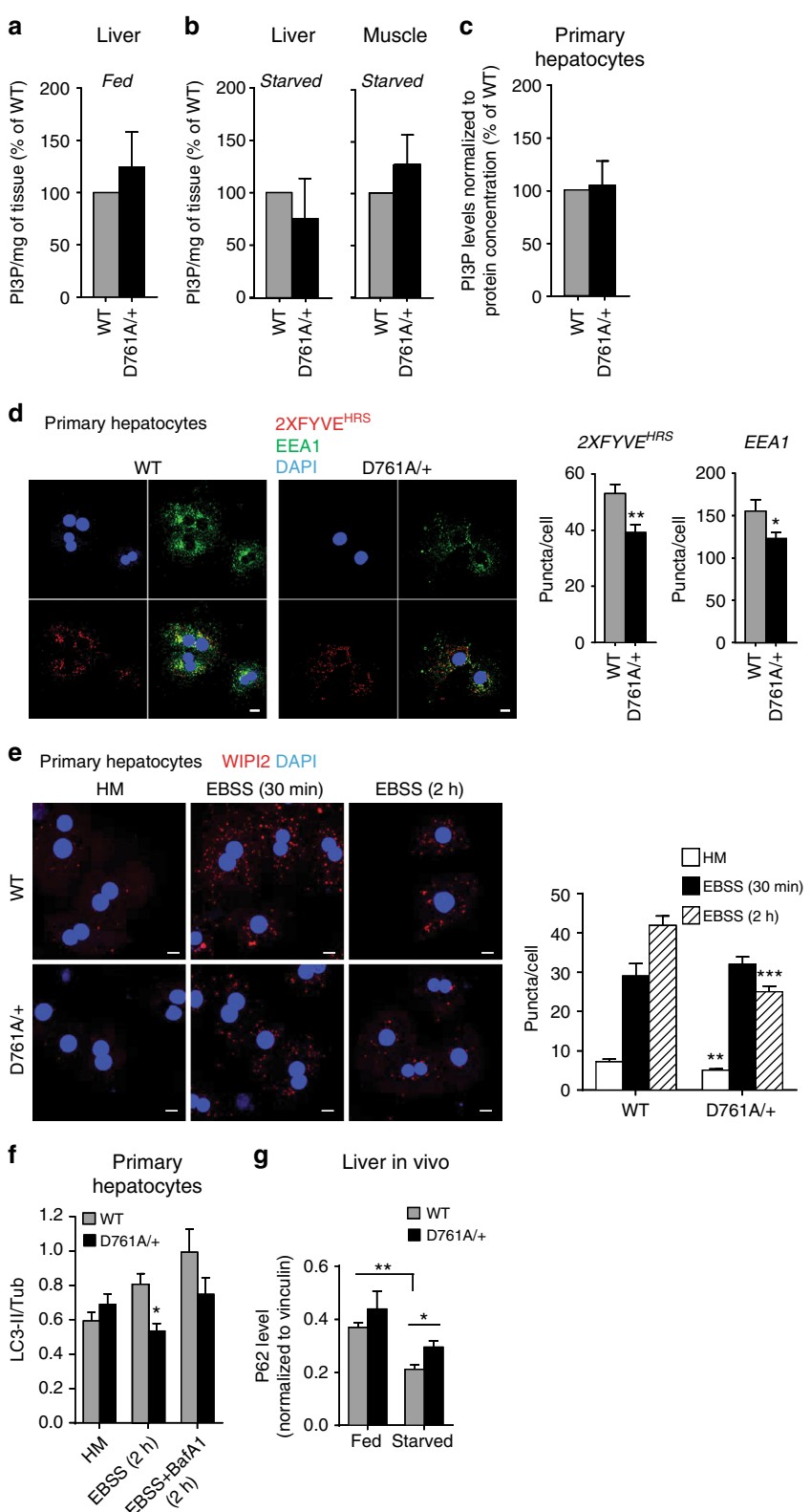

Fig. 7a, e, h) and hepatocytes (Fig. 9b; Supplementary Fig. 7i). Importantly, treatment with Vps34-IN1 of WT myotubes, primary hepatocytes or C2C12 myoblast cells, also increased the levels of pAMPK$^{T172}$ and pACC$^{S79}$ (Fig. 9b; Supplementary Fig. 7h–j), further validating our genetic model of Vps34 inactivation.

ACC is a rate-controlling enzyme in the synthesis of malonyl-CoA, a critical precursor for fatty acid biosynthesis and a potent inhibitor of mitochondrial fatty acid β-oxidation. Malonyl-CoA is an intermediate in fatty acid synthesis and an allosteric inhibitor of carnitine palmitoyltransferase 1, which regulates the transfer of long-chain acyl-CoAs from the cytosol into the mitochondria. AMPK-mediated phosphorylation of ACC on S79 inhibits its activity and therefore the upregulation of pACC$^{S79}$ levels observed in Vps34$^{D761A/+}$ cells might result in decreased lipogenesis and/or increased fatty acid oxidation. In line with this, gene expression levels of key players in β-oxidation showed that Vps34 inactivation in cells and tissues increased gene expression of the transcriptional activators PPARα and PGC-1α and their target genes including Cpt1b and Cpt1c (Fig. 9e, f), suggesting that Vps34 inactivation increases fatty acid oxidation in liver.

TBC1D1, a Rab GTPase-activating protein (GAP), and its homolog AS160 are critical for the translocation of the glucose transporter GLUT4 to the plasma membrane in skeletal muscle[32], the main tissue responsible for glucose uptake. Rab GAP activity is known to be inhibited by Akt and AMPK phosphorylation, leading to enhanced Rab-dependent translocation of GLUT4-containing vesicles to the plasma membrane[33]. In line with the increased basal levels of pTBC1D1 on T660 (Fig. 9a) and on S237 (Supplementary Fig. 7e) in Vps34$^{D761A/+}$ myotubes, these cells showed a significant increase in basal glucose uptake as compared to WT cells (Fig. 9c), a phenotype that could be mirrored by pharmacological inhibition of Vps34 in WT myotubes (Fig. 9d), with no further increases in glucose uptake seen upon insulin stimulation (Fig. 9c). The unaltered levels of the Akt phosphorylation site T642 in AS160 in Vps34$^{D761A/+}$ myotubes (Fig. 9a) suggest that Vps34 regulates GLUT4 translocation and glucose uptake in muscle cells via a pathway that does not involve the canonical insulin-driven Akt/AS160 pathway but instead acts through AMPK substrates such as TBC1D1.

To assess the involvement of AMPK in the increased insulin-independent increase in glucose disposal in Vps34-deficient muscle cells, blood glucose was assayed after an intraperitoneal injection of the cell-permeable AMPK activator 5-aminoimidazole-4-carboxamide ribonucleoside (AICAR). As shown in Fig. 9g, AICAR caused a significant reduction in blood glucose levels in WT mice, likely due to the combined effects of enhanced muscle glucose uptake activation and inhibition of hepatic glucose production. Interestingly, the AICAR-induced hypoglycaemic effect was similar between genotypes, but the blood glucose levels recovered significantly faster in Vps34$^{D761A/+}$ mice. This could be explained by a higher glucose disposal and/or a possible better hepatic glucagon response in Vps34$^{D761A/+}$ mice.

**Pharmacological Vps34 inhibition improves metabolism in vivo**. Our cell-based studies above showed that pharmacological inactivation of Vps34 by Vps34-IN1 mimicked the effect of genetic inactivation of Vps34. To assess the potential therapeutic relevance of our findings, we dosed HFD-fed mice with compound 19, a selective Vps34 kinase inhibitor whose in vivo pharmacology has been characterized[34]. Although compound 19 is a potent and selective Vps34 inhibitor and is able to inhibit autophagy in cellular and mouse models, its pharmacokinetic properties need to be improved for therapeutic use. Indeed, this is a first generation Vps34 inhibitor with a very short in vivo half-life ($t_{1/2} \sim 1.2$ h by i.v. injection). Given that our genetic model of Vps34 inactivation leads to a sustained low level of inhibition (50%), it is challenging to achieve a similar effect with the Vps34 inhibitors currently available in the public domain. However, we tested compound 19 in a proof-of-concept study, with a 5-week dosing (5 days on/2 days off drug) at a relatively low dose of 20 mg/kg (experimental design is shown in Supplementary Fig. 8a). This treatment regimen was well-tolerated, did not induce weight loss (Supplementary Fig. 8b) and improved glucose tolerance and insulin sensitivity (as assessed by GTT and ITT) after 2 weeks of drug treatment, with beneficial metabolic effects maintained after 5 weeks of treatment (Fig. 10a, b; Supplementary Fig. 8c, d). Moreover, treatment with compound 19 in vivo mirrored the signaling impact on pACC$^{S79}$ (Fig. 10c), further validating our observations of genetic inactivation of the kinase activity of Vps34.

## Discussion

In this study, we report that partial in vivo inactivation of the Vps34 isoform of PI3K enhances insulin sensitivity and glucose tolerance, identifying this kinase as a new drug target for the treatment of insulin resistance including Type-2 diabetes, a condition in which the unmet therapeutic need remains substantial. The mode of insulin sensitization by Vps34 inhibition has similarities to that of the current front-line anti-diabetic drug, Metformin, including alteration of cellular energy homeostasis but with a distinct primary mechanism of action. The distinct but overlapping mechanisms of action of Vps34 inhibitors and Metformin mean that a Vps34 inhibitor could have major clinical

**Fig. 5** Impact of Vps34 inactivation on cellular PI3P and autophagy. **a–c** Analysis of total PI3P levels by mass assay in different cell types/tissues. Mice were randomly fed or starved overnight. Hepatocytes were cultured overnight in insulin-free HM media. 5 mice/genotype were used for all tissues, except for primary hepatocytes (cell cultures derived from 4 individual mice/genotype). Data represent mean ± SEM (non-parametric Mann–Whitney t-test). **d** Left panel, representative images of confocal analysis of endogenous PI3P pools in primary hepatocytes. Cells were cultured overnight in insulin-free HM, digitonin-permeabilized and co-stained using a GST-2xFYVE$^{HRS}$ probe (red) or EEA1 antibody (green). DAPI-stained nuclei are shown in blue. Cell cultures derived from 3–5 independent mice/genotype were used. Scale bar, 20 μm. Right panel, quantification of FYVE and EEA1 puncta number from confocal images using Metamorph software. Cell cultures derived from 3–5 independent mice/genotype were used. Data represent mean ± SEM (Student t-test). **e** Left panel, representative images of confocal analysis of WIPI-2 puncta (red) of primary hepatocytes upon starvation (EBSS). Representative data from three independent experiments. DAPI-stained nuclei are shown in blue. HM: Hepatocyte medium + insulin. Scale bar, 20 μm. Right panel, Quantification of WIPI-2 puncta in primary hepatocytes from confocal images using Metamorph software. HM: Hepatocyte medium (with insulin). Hepatocyte cultures derived from 3–5 independent mice/genotype were used. Data represent mean ± SEM (Student t-test). **f** Quantification of the LC3 lipidation assay in primary hepatocytes shown in Supplementary Fig. 4b. Cells were freshly isolated from liver and were grown overnight in HM. For autophagy induction, cells were incubated in EBSS in absence or presence of 100 nM Bafilomycin A1 (BafA1) for the indicated times. Cell lysates were immunoblotted with the indicated antibodies. Representative data from three independent experiments are shown. Data represent mean ± SEM (Student t-test). **g** Quantification of p62 levels in liver tissues shown in Supplementary Fig. 4c. Fresh livers were snap-frozen and lysed before performing immunoblotting with the indicated antibodies. $n = 4$ mice/genotype. Data represent mean ± SEM (non-parametric Mann–Whitney t-test) *$p < 0.05$, **$p \leq 0.01$, ***$p \leq 0.001$

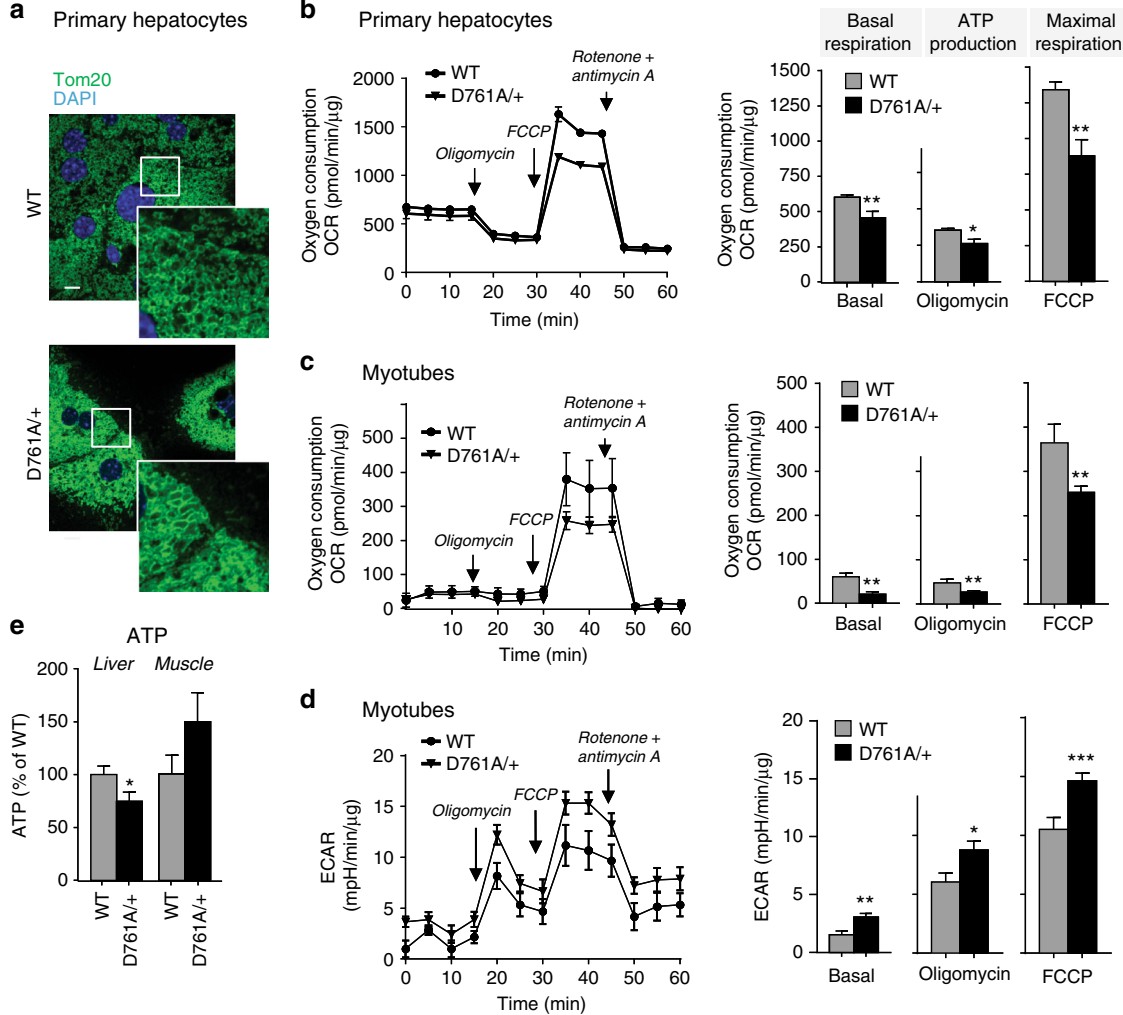

**Fig. 6** Impact of Vps34 inactivation on mitochondrial content and function. **a** Confocal images of Tom20 staining (mitochondrial marker) in primary hepatocytes in insulin-free HM media. DAPI-stained nuclei are shown in blue. A representative image of 50 cells analyzed is shown. Scale bar, 20 μm. **b**, **c** Seahorse XF24 measurement of oxygen consumption (OCR) by hepatocytes or differentiated myotubes isolated from WT and Vps34[D761A/+] mice. Quantification of basal respiration, ATP production and maximal respiratory capacity are shown on the graphs. Hepatocyte cultures derived from 3–5 independent mice/genotype; myotube cultures derived from three independent mice/genotype. Data represent mean ± SEM (Student t-test). **d** Seahorse XF24 measurement of extracellular acidification rate (ECAR) of differentiated myotubes. Myotube cultures were derived from three independent mice/genotype. **e** Total ATP levels in liver and gastrocnemius muscle in randomly fed mice. Data represent mean ± SEM (non-parametric Mann–Whitney t-test). 5 mice/genotype. *$p < 0.05$, **$p \leq 0.01$, ***$p \leq 0.001$

utility in the large population of diabetic patients where Metformin is contra-indicated or not tolerated, as discussed in more detail below.

To maintain metabolic homeostasis and viability, the cell must respond to changes in nutrient availability and ensuing energy stress. One of the key cellular responses to nutrient withdrawal is the upregulation of autophagy, an evolutionarily conserved process that degrades and recycles cytoplasmic components, such as dysfunctional proteins and organelles, in lysosomes[35]. The role of autophagy in metabolic adaptation at the organismal level is unknown, with conflicting evidence provided by mouse gene KO studies of components of the autophagy machinery. Indeed, given that gene KO of key autophagy-related genes (Atgs) leads to embryonic or neonatal lethality, tissue-specific conditional KO studies have been performed, revealing negative or positive metabolic impacts, depending on the gene and tissue targeted[36–41]. The metabolic impact of systemic autophagy deficiency therefore remains enigmatic and only few studies have provided insight into this question. These include mice with mosaic

deletion of Atg5[42] or mice with heterozygous KO of Beclin-1[43], which both display increased lipid accumulation in the liver[42, 43], with no changes in insulin sensitivity, glucose clearance and body weight in Beclin-1 heterozygous KO mice[44]. Here, we report that partial inactivation of Vps34 leads to a modest dampening of autophagy in the liver. This relatively weak impact might be due to (i) the partial inactivation of Vps34; (ii) cells switching to Vps34- and PI3P-independent autophagy[5, 45], and/or (iii) the increased activation of AMPK, a positive regulator of autophagy[46]. All these phenomena could mask the effect of Vps34 inactivation on autophagy, dampening possible negative organismal impacts and providing an acceptable therapeutic window. In general, it would be interesting to address whether heterozygosity for autophagy genes (Atgs) would lead to a systemic metabolic improvement at the organismal level. So far no phenotype has yet been reported in those settings.

Recent studies[47, 48] reported on tissue-specific deletion in mice of Vps15, an obligate binding partner of Vps34. Although liver-specific Vps15 KO leads to insulin sensitization[48], muscle-specific

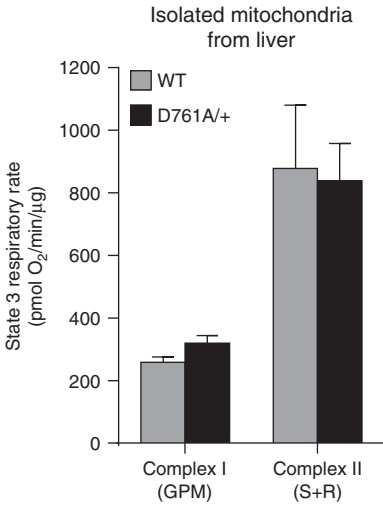

**Fig. 7** Impact of Vps34 inactivation on respiration of isolated mitochondria. State 3 respiratory rate of freshly isolated mitochondria from liver, driven either by complex I (glutamate/pyruvate/malate) or complex II (succinate + rotenone). Data represent mean ± SEM of three independent experiments

Vps15 KO did not affect whole-body glucose metabolism[47]. It is therefore not clear at present whether systemic abrogation of Vps15 expression would lead to metabolic improvement, and whether the observed effects in the liver-specific Vps15 KO are due to an impact on Vps34 itself. Indeed, in these Vps15 KO models, the expression of the components of Vsp34 complexes, including Vps34 itself, was severely reduced, complicating the interpretation of the observed phenotypes. The reported mechanism of insulin sensitization in Vps15 liver KO mice is a reduced endocytosis and degradation of the insulin receptor, resulting in enhanced insulin-mediated Akt signaling[48]. Although we observed a modest impact of Vps34 inactivation on the endosomal marker (EEA1) but no impact on Akt signaling, it is unlikely that insulin receptor trafficking plays a major role in the metabolic sensitization induced by Vps34 kinase inactivation.

Given its wide tissue distribution, the organismal impact of Vps34 inactivation is expected to be multifactorial (schematically summarized in Fig. 10d). The in vivo mechanism whereby the overall levels of blood glucose are reduced upon Vps34 inactivation is through a reduction in hepatic glucose production (gluconeogenesis) and an increase in glucose uptake in the muscle, with reduced autophagy most likely contributing to these effects. Indeed, although autophagy is tightly controlled by amino acid availability, amino acids produced by autophagy via its proteolytic function can be used for energy production (gluconeogenesis and ATP)[49, 50].

Lysosomes are the terminal degradative compartment of both the endocytic and autophagic pathways. Vps34 inhibition itself is known to disrupt lysosomal function[31, 51, 52], primarily through blocking sorting and trafficking of receptors and hydrolases on route to lysosomes[6, 52]. On the basis of this and on our findings that both endosomal (EEA1) and autophagic (p62, WIPI-2) markers were altered upon Vps34 inactivation, we speculate that the reduction in amino acids level results from a defect in lysosomal function. Our observations are in line with the notion that partial Vps34 inactivation reduces the proteolysis function of hepatic autophagy, thereby reducing the levels of specific gluconeogenic amino acids and reduced gluconeogenesis in Vps34[D761A/+] hepatic cells. A limited availability of substrates for the TCA cycle upon Vps34 inactivation could also explain the reduced mitochondrial respiration observed in Vps34[D761A/+]

hepatic cells, leading to a decrease in the levels of ATP and consequent activation of the AMPK energy sensing pathway. Activation of AMPK is known to stimulate mitochondrial biogenesis through upregulation of peroxisome proliferator-activated receptor-γ co-activator 1α (PGC-1α) activity. Such an increased AMPK-PGC-1α-mediated response in Vps34[D761A/+] hepatocytes may explain the ~20–25% increase in mitochondria content and increased expression of genes encoding mitochondrial proteins in these cells.

Although the impact of Vps34 inactivation on autophagy in the muscle remains to be determined, the increased levels of the AMPK-dependent phosphorylation of TBC1D1, an important regulator of glucose uptake, in Vps34[D761A/+] myotubes, strongly indicates that activation of AMPK underpins the reduced glucose uptake in muscle. Given that mitochondrial respiration was also reduced in Vps34[D761A/+] myotubes, we speculate that AMPK pathway activation in these cells results from a mechanism similar to that in Vps34[D761A/+] hepatocytes.

Metformin and the thiazolidinediones are the only two classes of insulin-sensitizing drugs that are currently available. Our data show that the mode of insulin sensitization by Vps34 inhibition has similarities to that of Metformin, the most frequently prescribed drug for Type-2 diabetes, but acts through a distinct primary mechanism of action. Indeed, whereas Vps34 inhibition indirectly inhibits mitochondrial respiration, via partial inactivation of autophagy and limiting substrate availability for respiration, Metformin directly disrupts mitochondrial function by inhibiting electron transport chain complex I[46]. Thus, both Vps34 inactivation and Metformin treatment alter cellular energy homeostasis and consequently activate the AMPK pathway, whose role in the action of Metformin still remains uncertain[53, 54]. Around 5% of diabetic patients do not tolerate Metformin, due mostly to gastro-intestinal side effects, such as nausea and/or diarrhea, whereas in many more patients where Metformin is effective it is withdrawn in the face of declining renal or cardiac function due to fears of lactic acidosis (see ref. [55] and http://www.fda.gov/Drugs/DrugSafety/ucm493244.htm). Although it is not expected that Vps34 inhibitors would replace Metformin, we expect this class of inhibitors to be complementary. Indeed, Vps34 inhibitors could have major clinical utility in the large population of diabetic patients where Metformin is contra-indicated or not tolerated, especially if Vps34 inhibitors would have a different side effect or pharmacokinetic profile to Metformin. Insulin resistance is closely associated with major diseases including Type-2 diabetes, the spectrum of fatty liver disease from steatosis through to cirrhosis, metabolic dyslipidaemia, ovulatory dysfunction and subfertility, and some cancers. Our data suggest that a potent and selective Vps34 inhibitor might offer a novel class of agent in the management of these clinical settings.

## Methods

**Mice.** Gene targeting was carried out by Taconic Artemis 467 (Cologne, Germany) and shown schematically in Supplementary Fig. 1a. All experiments were performed on 7- to 12-week-old male C57BL/6J mice, unless otherwise specified. Mice were kept on normal chow diet (20% protein, 75% carbohydrate, 5% fat) on a 12 h light-dark cycle (lights on at 7 a.m.) with free access to water in individually ventilated cages. Mice were cared for according to UK Home Office regulations, with procedures approved by the Ethics Committees of Queen Mary University London, UK and University College London, UK. For high-fat diet experiments, mice were maintained on diet 824,053 from Special Diet Services Inc. (20% protein, 35% carbohydrate, and 45% fat) for 16 weeks, or on diet 58Y1 (originally manufactured as "D12492") from Test Diet (18.1% protein, 20.3% carbohydrate, and 61.6% fat) for 7 weeks.

**Creation of Vps34[D761A] mice and genotyping.** Mouse gene targeting was performed by Artemis (Cologne, Germany) in C57BL/6NT embryonic stem cells. Mice were backcrossed on the C57BL/6J strain (Charles River) for >10 generations, and mice used for experiments were on C57BL/6J background, with WT littermates

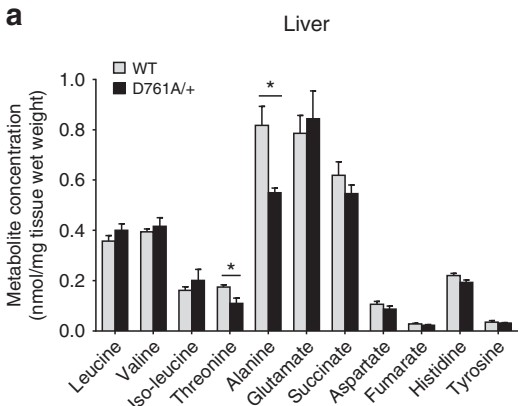
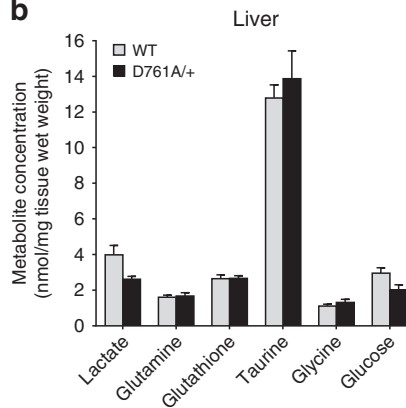

**Fig. 8** Metabolite concentration in fasted liver. **a, b** Livers from overnight-fasted mice were collected, snap-frozen and processed for NMR analysis. Data represent mean ± SEM (non-parametric Mann–Whitney t-test). 4–5 mice/genotype. *$p < 0.05$, **$p \leq 0.01$, ***$p \leq 0.001$

used as controls. The sequences of the primers used for genotyping are: forward primer (a in Supplementary Fig. 1a): 1450_33: GCTGGTAGTACTGATGTTGC; antisense primer (b in Supplementary Fig. 1a): 1450_34: GCATGGTCC-TACTTTCTTCC, with expected fragments of 327 bp (WT) and 444 bp (KI). PCR conditions were as follows: 95 °C for 5 min, 34 cycles of (95 °C for 30 s, 60 °C for 30 s, 72 °C for 1 min), and 72 °C for 10 min. The presence of the D761A mutation was verified by sequencing of a PCR fragment generated using forward primer: 1436_29: ATATGGTATCTCCTTCCTGG (c in Supplementary Fig. 1a) and anti-sense primer 1436_30: CACTGCTTTCGGAACTCTTG (d in Supplementary Fig. 1a) using the PCR conditions described above. PCR reactions were performed using Titanium Taq polymerase (Clontech) on a ThermalCycler (MJ Research).

**Antibodies and reagents**. All antibodies used in this study are listed in Supplementary Table 4. They were all reactive with mouse proteins as follows: EEA1, GST, pAkt (S473), pAkt (T308), pAMPK (T172), pACC (S79), pPRAS40 (T246), pTSC2 (T1462), pS6K (T389), pS6 (S240/44), pTBC1D1 (S660), pAS160 (T642), pGSK3 (S21/9), Vps34 (for western blot), Beclin-1 (Cell Signalling Technology); Vps34 (rabbit polyclonal antibody for immunoprecipitation for in vitro kinase assay; provided by J. Backer, New York); Tom20 (FL145; Santa Cruz); UVRAG (MBL); Vps15 (Epitomics); p62 (Novus Biologicals); vinculin, α-tubulin, Atg14/ Barkor (Sigma); LC3 (2G6; Nanotools); pTBC1D1 (S237) (Millipore); total OXPHOS cocktail (Abcam). Antibodies to WIPI-2 were provided by Sharon Tooze (Francis Crick Institute, UK). Mitotracker FM green was from Molecular Probes. Unless otherwise mentioned, phosphate-buffered saline (PBS; Sigma) was $Ca^{2+}$- and $Mg^{2+}$-free. All culture media for primary cell culture were from Invitrogen. A plasmid expressing GST-2xFYVE[HRS][28] was provided by Harald Stenmark, Norway. Recombinant GST protein was purified from E. coli BL21(DE3) cells according to the manufacturer's instructions. All buffers used during purification of the GST-fusion protein were EDTA-free and the recombinant protein was dialyzed against HEPES buffer pH 7.4 containing 10 μM $ZnCl_2$. FCCP, oligomycin and antimycin A were purchased from Sigma-Aldrich. Rotenone was purchased from MP Biomedicals (Santa Ana, CA). Agonists used were human (Actrapid) and bovine (Sigma) insulin for in vivo and in vitro experiments, respectively. Bafilomycin A1 was from Sigma.

**Hepatocyte isolation and culture**. Cells were cultured in a humidified incubator at 37 °C and 5% $CO_2$. Primary mouse hepatocytes were isolated from 8- to 12-week-old mice as described, with minor changes[56]. Briefly, primary hepatocytes were isolated by a two-step perfusion protocol using collagenase I (Sigma) and seeded on collagen-coated plates in William's E GlutaMAX medium containing 0.1% BSA, 1% penicillin/streptomycin, 25 nM dexamethasone (Sigma) and 680 nM insulin (further referred to as Hepatocyte Medium (HM)), further supplemented with 10% (v/v) FBS. After 4 h incubation at 37 °C to allow cell adhesion, the medium was replaced by HM. For insulin stimulation studies, HM was replaced by HM without insulin overnight, followed by addition of 100 nM insulin for the indicated times. In some experiments, the Vps34-selective inhibitor Vps34-IN1[30] was added 18 h before treatment. For autophagy studies, HM was removed and cultures were washed twice with HM or with amino acid- and insulin-free medium (EBSS; Invitrogen) and cells maintained in EBSS for 0.5 or 2 h.

**Myoblast isolation, culture and myotube differentiation**. Cells were cultured in a humidified incubator at 37 °C and 5% $CO_2$. Primary muscle cells (myoblasts) were obtained from Gastrocnemius and Tibialis anterior muscle from 3- to 4-week-old mice as described[57]. In brief, muscles were partly digested with four sequential 10 min incubations in DMEM/F12 + GlutaMAX-1 medium containing 0.14% pronase (Sigma, P8811). The supernatants from the second, third and fourth

digestions were pooled and filtered through a 100-μm cell strainer. Cells were centrifuged, washed twice, counted and plated at low density (100 cells/$cm^2$) in 12-well plates coated with gelatin (Sigma; G1393). Cells were grown in complete medium, composed as follows: DMEM/F12 + GlutaMAX-1 (Gibco), recombinant human FGF (2.5 ng/ml final; R&D Systems), 20% fetal bovine serum (Gibco), 50 U/ml penicillin/streptomycin. After 1 week, wells containing myoblasts without contaminating fibroblasts were trypsinized, pooled and passaged. Complete medium was changed every 2 days, and cultures were trypsinized before subconfluency to avoid differentiation. To differentiate into myotubes, myoblasts were plated on Matrigel-coated dishes at $3 \times 10^4$ cells/$cm^2$ (Corning, diluted 1/10 in DMEM/F12 + GlutaMAX-1). After 6 h, cells were switched to differentiation medium (DMEM/ F12 + GlutaMAX-1 containing 2% horse serum). The medium was changed twice during differentiation. Cell fusion and differentiation into multinucleated myotubes were monitored using phase contrast microscopy. All experiments on myotubes were performed after 7 days of differentiation.

**Lipid kinase assay**. Lipid kinase assay on Vps34 immunoprecipitates using PI as a substrate was performed as described[58] with minor changes. Briefly, cells or tissues were lysed in lysis buffer (LB) (1% Triton X-100, 150 mM NaCl, 50 mM Tris pH 7.4, 10% Glycerol, 1 mM $CaCl_2$, 1 mM $MgCl_2$, Protease/Phosphatase inhibitors from Merck) and incubated for 25 min on ice. Lysates were spun at 15,000 rpm for 10 min at 4 °C. Vps34 immunoprecipitation was performed using 1 mg of total protein and protein A Sepharose (Amersham-17-0469-01) and Vps34 antibody (rabbit polyclonal antibody for immunoprecipitation kindly provided by J. Backer, New York) at 4 °C for at least 2 h. Beads were resuspended in kinase buffer (20 mM Tris, pH 7; 67 mM NaCl; 10 mM $MnCl_2$; 0.02% (w/v) NP-40). Kinase assay was performed using 0.1 μCi/μl radioactive labeled γ-ATP (32 P) (Hartmann Analytic #SRP 401) per reaction for 15 min at 30 °C. Extracted lipids (using Choloroform-Methanol extraction protocol) were separated using a Silica 60 thin layer chromatography (TLC) plate and PI3P spots were quantified using Typhoon Imaging System (GE Healthcare).

**Western blot analysis**. Tissues were lysed in 20 mM Tris.HCl pH 8.0, 1% NP-40, 5% glycerol, 138 mM NaCl, 2.7 mM KCl, 20 mM NaF, 5 mM EDTA, and Protease/ Phosphatase inhibitors cocktail from Merck. To remove cell debris, homogenates were spun at 13,000 rpm for 10 min at 4 °C and the supernatant fraction recovered. Protein concentration was determined by colorimetric assay (Bradford assay, Biorad). Protein extracts were resolved by SDS-PAGE, transferred to PVDF membranes and incubated overnight at 4 °C with specific antibodies. Antigen-specific binding of antibodies was visualized by ECL. Uncropped immunoblots and larger blot areas of the main figures are shown in Supplementary Fig. 9.

**Immunofluorescence**. Hepatocytes were seeded at $2.5 \times 10^5$ per well on collagen-coated glass coverslips in 6-well plates as described above. Cells were fixed with 4% paraformaldehyde and permeabilized for 10 min with 0.2% Triton X-100. Permeabilized cells were blocked in PBS/2% BSA for 1 h and incubated in PBS/2% BSA with the indicated antibodies at 4 °C overnight. After three washes with PBS, cells were incubated with species-specific Cy3- or FITC-labeled secondary antibodies for 1 h at room temperature. After three washes with PBS, coverslips were mounted on glass slides using Vectashield containing DAPI (Vector Laboratories). Staining with GST-2xFYVE[HRS] was performed as described[28, 59] using permeabilization with digitonin as described. Briefly, after treatment cells were fixed in 4% PFA before being washed in PBS/2% BSA for 5 min, followed by permeabilization with 20 μM digitonin in PBS/2%BSA for 5 min at room temperature, followed by three washes in PBS/2% BSA. Cells were incubated with the GST-2xFYVE[HRS] probe (0.5 μg/ml) for 30 min in PBS/2% BSA. After three washes with PBS/2% BSA,

the cells were incubated with antibody to GST for 45 min, with further incubation with secondary antibody reagents as described above. Immunostaining for autophagy markers (LC3 and WIPI-2) was performed as follows: cells were washed twice with PBS before adding cold (−20 °C) methanol for 15 min. Cells were then washed twice with PBS and blocked in PBS/3% BSA for 1 h and incubated overnight at 4 °C in PBS/3% BSA with the indicated antibodies. After three washes with PBS, cells were incubated with species-specific Cy3- or FITC-labeled secondary antibodies for 1 h at room temperature. After three washes with PBS, the coverslips

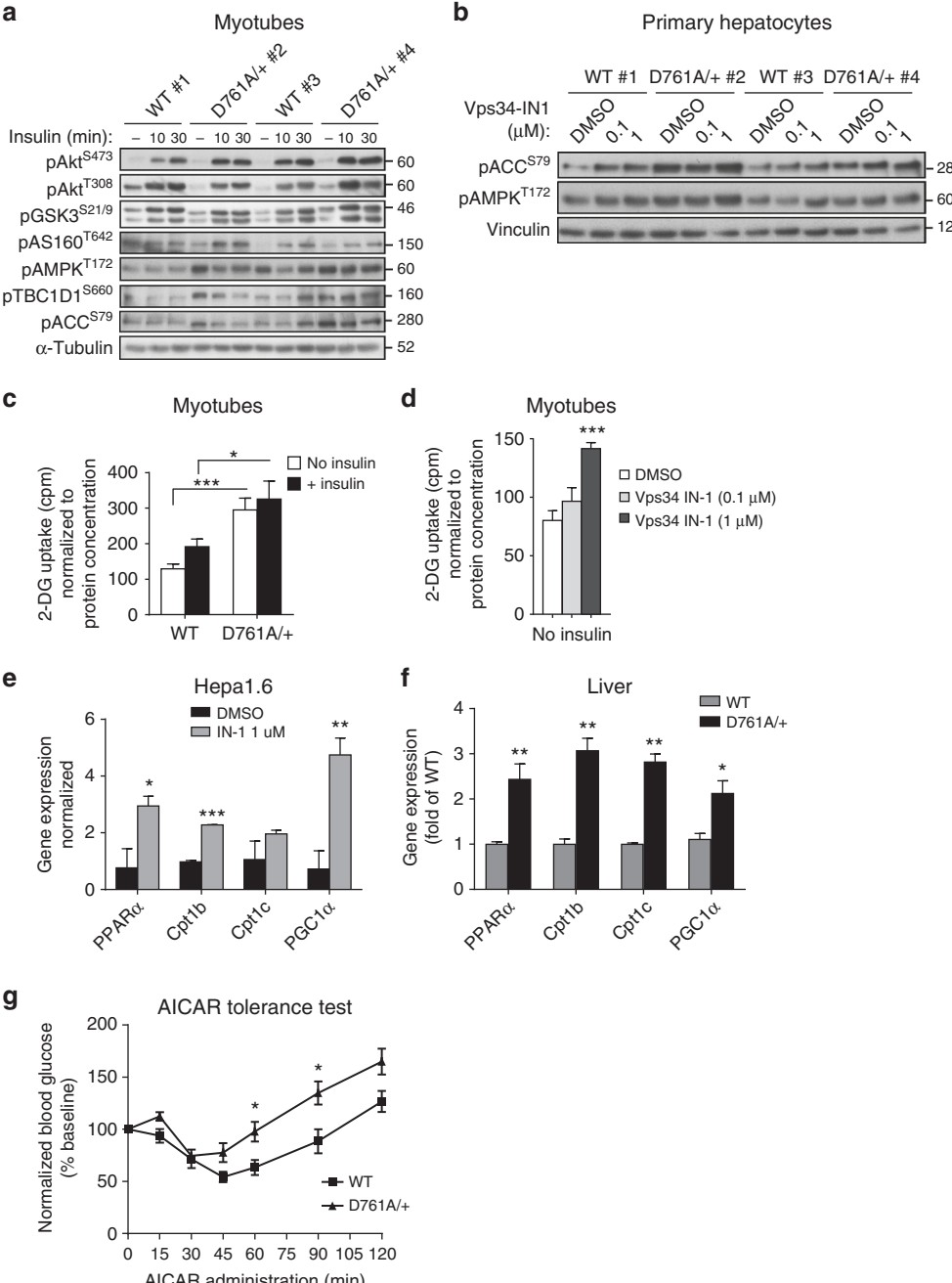

**Fig. 9** Impact of Vps34 inactivation on insulin-mediated Akt/mTORC1 and AMPK signaling. **a** Cultured myotubes were starved for 5 h and stimulated for the indicated time points with 100 nM insulin. Cell lysates were immunoblotted with the indicated antibodies. Representative data from three independent experiments are shown. **b** Primary hepatocytes were cultured overnight in insulin-free HM in the presence of DMSO or Vps34-IN1 at the indicated doses. Cell lysates were immunoblotted with the indicated antibodies. Representative data of three independent experiments are shown. **c** 2-Deoxy-ᴅ-glucose uptake in differentiated myotubes. Myoblasts were differentiated for 6–7 days, starved for 5 h before and stimulated with 100 nM insulin for 20 min. Myotube cultures derived from three independent mice/genotype. Data represent mean ± SEM (Student $t$-test). **d** 2-Deoxy-ᴅ-glucose uptake in differentiated myotubes. Myoblasts were differentiated for 6–7 days, cultured overnight in DMSO or in presence of 1 μM Vps34-IN1, then starved for 5 h before the assay. Myotube cultures were derived from three independent mice/genotype. Data represent mean ± SEM (Student $t$-test). **e, f** Expression levels of genes related to fatty acid β-oxidation in Hepa1.6 cells. **e** treated overnight in absence or in presence of 1 μM Vps34-IN1 in serum-free media and in liver tissues **f** from WT and Vps34$^{D761A/+}$ starved mice. Expression levels were quantified by RT-qPCR from independent cell cultures and from $n = 5$ mice/genotype. Data represent mean ± SEM (Student $t$-test). **g** AICAR tolerance test was performed by intraperitoneal injection of 0.15 g AICAR per kg body weight into overnight-fasted 9–11-week-old mice. ≥ 5–11 mice/genotype were used. Data represent mean ± SEM (non-parametric Mann–Whitney $t$-test). *$p < 0.05$, **$p < 0.01$, ***$p < 0.001$

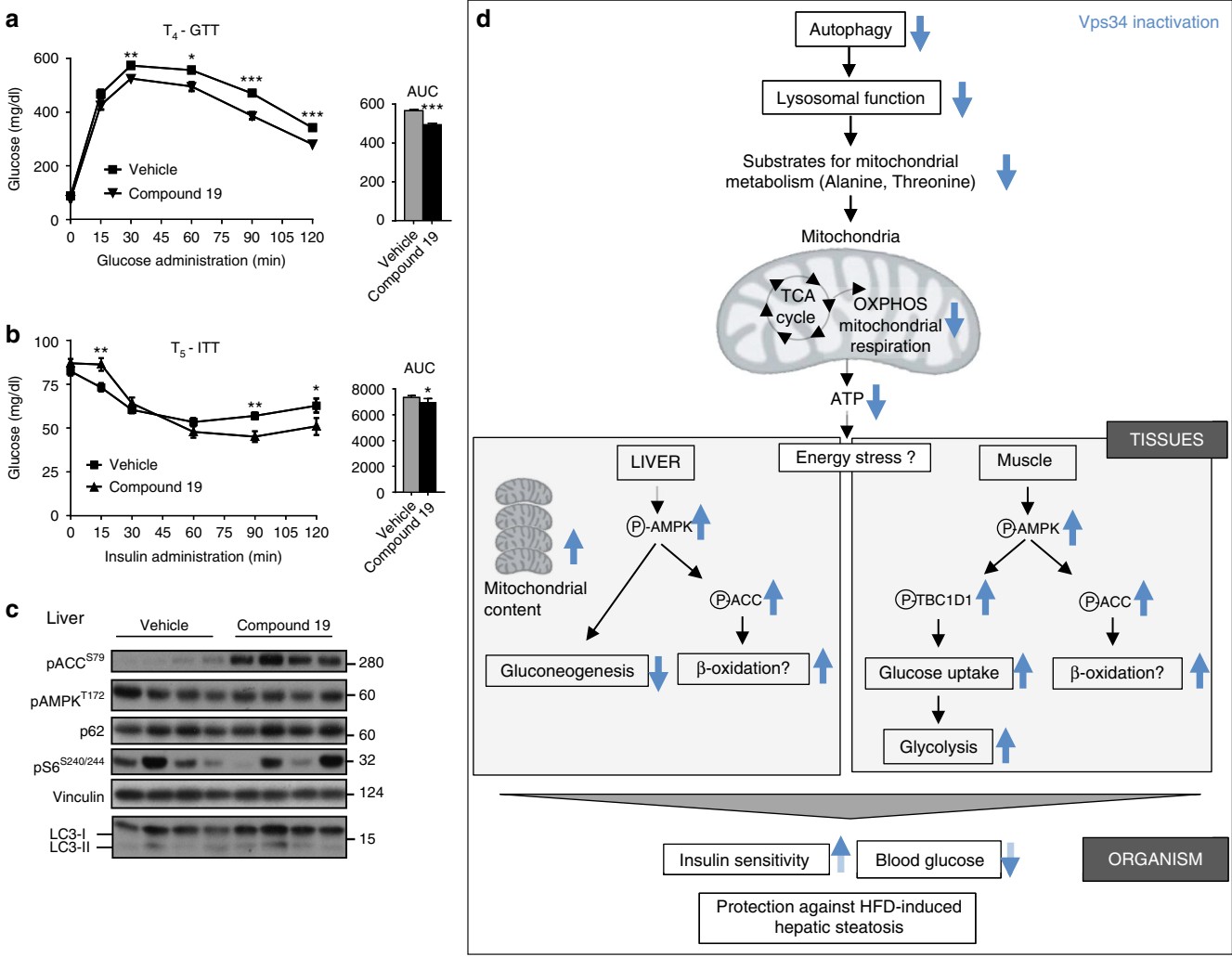

**Fig. 10** Impact of pharmacological inhibition of Vps34 on insulin resistance in 60% HFD-fed mice. **a** GTT after intraperitoneal injection of 2 g/kg of glucose after overnight starvation in 60% HFD-fed mice treated with compound 19 for 4 weeks. Data represent mean ± SEM (non-parametric Mann–Whitney $t$-test). 10–12 mice/genotype. $*p < 0.05$, $**p < 0.01$, $***p < 0.001$. **b** ITT after intraperitoneal injection of 0.75 U/kg of insulin after overnight starvation in 60% HFD-fed mice treated with compound 19 for 5 weeks. Area under the curve (AUC) is shown adjacent to the graph. Data represent mean ± SEM (non-parametric Mann–Whitney $t$-test). 10–12 mice/genotype. $p < 0.05$, $**p < 0.01$, $***p < 0.001$. **c** Tissue homogenates from liver isolated from overnight-starved mice were immunoblotted using the indicated antibodies. Each lane represents an individual mouse. 60 μg of protein was loaded per lane. **d** Schematic representation of the metabolic impact of partial systemic Vps34 inactivation, as described in the manuscript. See text for details

were mounted on glass slides as described above. All coverslips were analyzed using a ×63 objective on a 710 Zeiss confocal microscope.

**MitoTracker green FM staining and flow cytometry**. Hepatocytes were seeded at a concentration of $2 \times 10^5$ cells/ml and cultured overnight in HM. Following one wash in PBS, cells were trypsinized, pelleted and resuspended in HM containing 150 nM MitoTracker Green FM (Molecular Probes, Life Technologies) and incubated in the dark at 37 °C for 30 min. Cells were washed once and resuspended in FACS buffer (PBS + 1% FBS) containing 0.5 μg/ml DAPI before analysis on a FACScan flow cytometer (Becton Dickinson, San Jose, CA). Data were analyzed using FlowJo 8.6 software (Tree Star Inc., Ashland, OR).

**Metabolic analysis**. For glucose tolerance tests, mice were fasted overnight (16 h) followed by an intraperitoneal injection of 2 g glucose (20% solution; Baxter)/kg body weight. Blood glucose levels were monitored before and 15, 30, 60, 90, and 120 min after injection using blood collected from the tail vein using a Glucotrend glucometer (Roche Diagnostics). For insulin tolerance tests, mice were fasted overnight (16 h), followed by injection with human insulin (0.75 U/kg body weight). Blood from tail was collected before and 15, 30, 60, 90, and 120 min after injection and glucose levels were determined as described above. For in vivo insulin stimulation, mice were fasted overnight (16 h) followed by intraperitoneal injection of insulin (0.75 U/kg body weight) or vehicle (PBS). After 30 min, mice were sacrificed and tissues snap-frozen in liquid nitrogen. Triglyceride levels in liver

tissue were determined as described[60]. Briefly, 50 mg of liver tissue were homogenized in 900 ml of a 2:1 chloroform:methanol solution. Three hundred ml of methanol were added to the liver homogenate and vortexed followed by centrifugation for 15 min at 3000 rpm. The supernatant (412.5 ml) was transferred to a new glass tube and 200 ml of chloroform and 137.5 ml of 0.73% NaCl was added, and vortexed for 30 s. After centrifugation at 5000 rpm for 3 min, 400 ml of a 3:48:47 solution of chloroform:methanol:NaCl (0.58%) was added to the lower phase followed by another centrifugation at 5000 rpm for 3 min. The lower phase was washed three times, evaporated and resuspended in 1 ml of isopropanol. Triglyceride levels were measured using a standard assay kit (Infinity Triglycerides Reagent TR22421) from Thermo Scientific following manufacturer's instructions. Liver triglycerides were normalized by liver tissue weight. Serum levels of insulin, leptin, triglyceride, cholesterol, and adiponectin were measured using ELISA and colorimetric kits (Crystal Chem Inc. for insulin, Millipore for leptin and adiponectin; Cayman Chemical Company for triglyceride and cholesterol). Measurements of food intake were obtained with a CLAMS (Columbus Instruments) open-circuit indirect calorimetry system. Body composition was determined by magnetic resonance spectroscopy using an ECHO MRS instrument (Echo Medical Systems).

**Dosing of mice with compound 19**. Mice (10 mice per group) were subjected to 60% HFD for a period of 7 weeks. Two weeks into the HFD, mice were dosed with either vehicle or compound 19 (PO, Q.D. at 20 mg/kg of mouse weight) for five consecutive days a week (Monday to Friday). Starting from 2 weeks later, mice were starved overnight followed by a GTT ("T2-GTT" and "T4-GTT") and an ITT

("T3-ITT" and ("T5-ITT") every week, with a week of recuperation between the assays while still on drug. The body weight of mice was measured every day during the experiment. Compound 19[34] was dissolved in polyethylene glycol (20% v/v) and sonicated for 30 min until dissolved. Cremophor EL (5% v/v) was then added and vortexed. Citrate buffer (pH 4, 75% v/v) was then added and pH adjusted to 4.5. The reformulated compound was then aliquoted and stored at 4 °C for the whole course of the study (5 weeks). A fresh aliquot of reconstituted compound 19 was used each day. Mice were orally gavaged with vehicle or compound 19 at 20 mg/kg for 5 weeks with a cycle of five consecutive days on-drug and 2 days off-drug.

**Data availability**. All relevant data are available from the authors.

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

## Acknowledgements

We thank members from the Cell Signalling laboratory for discussions, Harald Stenmark (University of Oslo, Norway) for providing reagents, staff at TaconicArtemis (Cologne; Germany) for mouse gene targeting, Guillaume Halet (Rennes, France) for help with the blastocyst outgrowths, Remi Mounier (Lyon, France) for help with myoblast isolation and culture. and Zuncai Wang, Andrew Powers, Bryan Lafitte, and Brian Ethell (Novartis Institutes for BioMedical Research) for experiments to guide the in vivo use of compound 19. Postdoctoral fellowships were from EU Marie Curie (PIEF-GA-2009–252916) and EMBO (ALTF 753–2010) for SA and EU Marie Curie (PIIF-GA-2009–252846) for C.C. J. M.H. was a recipient of a doctoral fellowship from Eisai UK Ltd. Work in our laboratories was supported as follows: BV: MRC [G0700755], BBSRC (BB/I007806/1 and BB/M013278/1), CRUK (C23338/A15965), the Ludwig Institute for Cancer Research and the National Institute for Health Research (NIHR) UCL Hospitals Biomedical Research Centre; J.M.B.: NIH AG039632, GM112524. and the Albert Einstein Diabetes Research and Training Center Animal Physiology Core DK020541; E.G.: Barry Reed Cancer Research fund; G.S.: BBSRC (BB/L020874/1) and B.H.F.; S.S.: Anatomical Society of Great Britain (GT) and a Wellcome Trust Career Development Fellowship 074246/Z04/Z (S.S.); R.K.S.: Wellcome Trust (WT098498) and M.R.C. (MRC_MC_UU_12012/5); S.A. T. and L.C.: the Francis Crick Institute, which receives its core funding from CRUK (FC001187), MRC (FC001187), and the Wellcome Trust (FC001187); Y.-L.C.: the CRUK Cancer Imaging Centre in association with the MRC and DoH (England) grant C1060/A10334, C1060/A16464, NHS funding to the NIHR BRC; B.P.: Inserm and the Fondation pour la recherche médicale. B.P. is a scholar of the Institut Universitaire de France.

## Author contributions

B.B., S.A., W.P., D.M., Y.-L.C., G.C., C.V., J.M.H., P.J.V., L.C., C.P., K.A., E.G., V.R., G.T., S.S., C.C., and R.S.S. performed experiments and data analysis, with input from G.S., M. A.W., J.M.B., R.K.S., B.P., S.T., and B.V.; C.S. performed and interpreted histopathology. E.P.K., L.O.M., R.K.S., B.B., and B.V. designed and interpreted the experiments using compound 19. B.B., M.A.W., and B.V. wrote the paper.

## Additional information

**Competing interests:** B.V. is consultant to Karus Therapeutics (Oxford, UK). L.M. was an employee of Novartis who has developed vps34 inhibitors. The remaining authors declare no competing financial interests.

