## [Peer Review File · Nature Communications]

Reviewers' comments:

Reviewer #1 (Remarks to the Author):

To investigate the role that Vps34 PI3K, a key lipid kinase known as autophagy initiator, plays on organismal metabolism at systemic levels, the authors generated a kinase-inactive (D761A) knock-in (KI) mouse model. Because they found that homozygous Vps34 D761A/ D761A KI mice are lethal at embryonic stage, they instead characterized heterozygous Vps34 D761A/ + KI mice. Assuming that Vps34 D761A/ + would cause 50% loss of kinase activity, the authors anticipated that this may better mimic the physiological/pharmacological effects of Vps34 inhibitor (at systemic levels). In the current manuscript, they report that Vps34 D761A/ + KI mice display enhanced glucose tolerance and insulin sensitivity, which was associated with partially dampened hepatic autophagy, altered metabolic switching and increased glucose uptake in muscle. Mechanistically, they found that even though Vps34-deficiency does not affect Akt signaling, it promoted activation of AMPK pathway (leading to insulin-independent glucose uptake in muscle). Collectively, they propose that systemic inhibition of Vps34 (by pharmacological means) would be beneficial as potential treatment of metabolic disorders.

General comments:

The authors used an elegant and appropriate model to ask a key biological/physiological question. As clearly described, Vps34 exists as multi-protein complex and knocking out Vps34 is expected to alter expression of its binding proteins, which may cause unexpected secondary effects. Given that here the authors' intention is to inhibit Vps34 activity like pharmacological intervention with complex intact, kinase-inactive KI is the right approach to take. The authors report some interesting metabolic phenotype of the Vps34 Het KI mice, but there are multiple fundamental issues (as described below) and data not compelling enough to support their conclusion.

Major comments:

1. The authors claim that Vps34 Het KI mice show enhanced insulin sensitivity, but (as described below) there is no clear evidence supporting this notion. Please clarify what is the definition of insulin sensitivity in the authors' point of view.

a. Based on the ITT results shown in Fig. 2D, the authors concluded that KI mice are more insulin sensitive compared to control animals. However, if you actually look at the data, KI mice have lower basal glucose, but upon inulin injection, kinetic (time-course) and magnitude of changes in blood glucose levels look nearly identical between control and KI mice. Please show the data as delta (%) changes from the baseline of respective genotype. If the glucose curve is comparable between control and KI mice, do the authors still claim KI mice are insulin sensitive? If so, please justify.

b. Assuming that insulin sensitivity is similar between control and KI (based on delta % change data), I will need to question the authors' interpretation regarding GTT. What is the potential reason that the KI animals are more glucose tolerant (assuming insulin sensitivity is similar between genotypes as argued above)? Please show that data 1st phase insulin secretion (plasma insulin levels following 5-10 min glucose injection) and/or show muscle/liver p-Akt blots. If the authors claim that basal AMPK levels might be increased and this explains higher glucose disposal per se, please show p-AMPK/p-ACC blots. It would be also informative if the authors perform AICAR tolerance test and to see if the KI mice show blunted response (blood glucose kinetic).

c. Fig. 4: Results of hyperinsulinemic-euglycemic clamp are not interpretable. As the authors do not show validation of "clamp" and also other associated key parameters. Please show (in Supplementary section): 1) glucose kinetics to demonstrate that glucose levels are clamped at euglycemic levels and there is no difference between the genotypes, 2) show glucose infusion rate and whole body glucose disposal data (Rd) as well.

2. The authors show that AMPK phosphorylation is increased in myotubes/hepatocytes from Vps34 Het KI mice/ Vps34 inhibitor-treated cells, and they claim this would be the key mechanism by which Vps34-deficiency/inhibition causes enhanced insulin sensitivity and protection from high fat diet induced embolic abnormalities. However, there are several questionable observations/interpretations that need to be addressed.

a. What is the mechanism by which AMPK is activated via Vps34-deficiency/inhibition? The authors show there is no change in ATP content in muscle/liver of Vps34 KI mice. Would this be different if authors measure ATP (AMP) levels in myotubes/hepatocytes (due to reduced mitochondria respiratory capacity) from Vps34 KI mice? If so, how do the authors reconcile in vivo and in vitro data?

b. The authors suggest that enhanced glucose uptake in Vps34-deficient muscle cells is mediated via AMPK-TBC1D1 axis and show increased p-S660 TBC1D1 signal. There is no evidence that this site has any functional significance. Other site S237/1 has been better characterized with some genetic evidence associated (O'Neill et al, PNAS, 2011, Frosig, J Physiol, 2010, Chen et al, Diabetologia, 2017).

c. Fig 9c and d: Vps34-deficient muscle cells have higher basal AMPK activity and insulin-independent 2DG uptake. They also show that Vps34 inhibitor causes an increase in 2DG in the absence of insulin. Based on these data with enhanced AMPK signaling coupled, they conclude that enhanced insulin-independent glucose uptake in muscle is likely AMPK-dependent via Vps34 inhibition. If that is the case, Vps34 inhibitor-mediated increase in 2DG should be abolished in Vps34-deficient muscle cells. It might be the case that Vps34 is simply a mitochondrial poison which indirectly activates AMPK independently of Vps34. Moreover, to verify that insulin-independent increase in glucose uptake in Vps34-deficient muscle cells is due to AMPK (Fig. 9c), please include AICAR/A769662 and show there is no or blunted increase by these activators (alternatively AMPK α 1/ α 2 can be knocked down by RNAi if feasible in the authors lab).

Reviewer #2 (Remarks to the Author):

The study evaluates a whole body heterozygous loss-of-function Vps34 mutant regarding a number of metabolic parameters. Mutants presented increased glucose tolerance and reduced predisposition for diet-induced hepatic steatosis. The phenotype was mostly due to reduced hepatic autophagy, which limited substrate availability of mitochondrial respiration and gluconeogenesis. Some of the metabolic outcomes were reproduced with a pharmacological inhibitor of Vps34. In general this is a fine work providing important advance in the field and identifying a potentially interesting target for the treatment of diabetes. Because of the potential impact of these findings, some additional experiments and controls are required.

Major comments

1. One major concern about the outcomes of partially inhibiting Vps34 is that mitochondria function is considerably modified. This could have a short term beneficial effect; however, in the long run this could severely impact on metabolic fitness. It is very important that ageing mutants are analyzed for: adiposity, blood lipids, fat deposition in blood vessels and muscle and markers of oxidative stress. Also, longevity should be determined (a simple Kaplan-Meier graph would be fine).
2. As repeatedly recommended in the series of guidelines for methods in autophagy research, transmission electron microscopy (TEM) is the gold-standard method for evaluation of autophagosome (Klionski et al Autophagy 2016 12:1-222). Critical experiments evaluating the impact of loss-of-function mutation of Vps34 should include TEM.
3. The kinase activity shown in figure 1b should include a positive control, another PI3K isoform.
4. Why in figure 2b fasting glucose is about 80 mg/dl for both WT and mutant, whereas in the

other experiments (Fig 2a and 2d) glucose is different between WT and mutant?

5. Authors should calculate kITT for the experiment shown in figure 2d. for stating that mutants are more sensitive to insulin glucose decay during ITT is more important than actual glucose levels.

Minor

1. In abstract (line 51) and discussion (line 370) authors propose that drugs that inhibit Vps34 could be potentially useful in patients that are intolerant to metformin. They say that about 10 % of patients are intolerant to metformin but no reference is presented. In fact, most patients (in my experience much more than 90%) tolerate metformin and the only real major problem associated with the use of this drug is lactic acidosis, which is an extremely rare event. Please consider reviewing this information.

2. It seems there is a splice in the bottom immunoblots for LC3 in Supplementary Figure 4. If that is the case, the splice should be clearly labeled or the experiment repeated in order to obtain a satisfactory image in a single gel.

Reviewer #3 (Remarks to the Author):

In this study Bilanges and coworkers addresses the impact of Vps34 (Vps34 phosphoinositide 3-kinase (PI3K)), a central player of autophagy, on energy homeostasis. The authors report that systemic Vps34 inactivation in mice leads to enhanced insulin sensitivity and improved glucose tolerance, with reduced hepatic glucose production and increased glucose uptake in muscle. At the mechanistic level, Vps34 inactivation dampened hepatic autophagy, limiting substrate availability for mitochondrial respiration and gluconeogenesis. In muscle, Vps34 inactivation triggers a metabolic switch from oxidative phosphorylation towards glycolysis and enhanced glucose uptake. Interestingly, Vps34 inactivation did not affect Akt signaling but activated the AMPK pathway, suggesting potential clinical utility for Vps34 inhibitors in Type-2 diabetic patients. This a very well performed study with potential therapeutic outcomes.

Major points

1. In Figure 2, the ITT between the two genotypes seems rather similar. How would it look expressed in percent of base line? The area under the curve should be measured.

2. Figure 3. It is surprising that insulin concentrations were not increased in response to HFD (i.e fasting hyperinsulinemia ?). In the same line, it is surprising not to have any increase in liver weight upon HFD treatment.

3. The observation that circulating levels of adiponectin is interesting. What is the

4. Glucose uptake is increased in skeletal muscle, with a only a tendency in white and brown adipose tissue. To exclude the participation of adipose tissues to the phenotype, insulin signaling experiments (such as Akt, IRS1/2 phosphorylation assays etc..) should be performed in white and brown adipose tissues.,and compared to muscle and liver.

5. Key genes of the β -oxidation pathway should be measured (PPAR α , LCPT1 etc..). Does increased in β -oxidation rates explain the lack of steatosis in response to HFD diet ?

Reviewer #1 (Remarks to the Author):

To investigate the role that Vps34 PI3K, a key lipid kinase known as autophagy initiator, plays on organismal metabolism at systemic levels, the authors generated a kinase-inactive (D761A) knock-in (KI) mouse model. Because they found that homozygous Vps34^{D761A/D761A} KI mice are lethal at embryonic stage, they instead characterized heterozygous Vps34^{D761A/+} KI mice. Assuming that Vps34^{D761A/+} would cause 50% loss of kinase activity, the authors anticipated that this may better mimic the physiological/pharmacological effects of Vps34 inhibitor (at systemic levels). In the current manuscript, they report that Vps34^{D761A/+} KI mice display enhanced glucose tolerance and insulin sensitivity, which was associated with partially dampened hepatic autophagy, altered metabolic switching and increased glucose uptake in muscle. Mechanistically, they found that even though Vps34-deficiency does not affect Akt signaling, it promoted activation of AMPK pathway (leading to insulin-independent glucose uptake in muscle). Collectively, they propose that systemic inhibition of Vps34 (by pharmacological means) would be beneficial as potential treatment of metabolic disorders.

General comments:

The authors used an elegant and appropriate model to ask a key biological/physiological question. As clearly described, Vps34 exists as multi-protein complex and knocking out Vps34 is expected to alter expression of its binding proteins, which may cause unexpected secondary effects. Given that here the authors' intention is to inhibit Vps34 activity like pharmacological intervention with complex intact, kinase-inactive KI is the right approach to take. The authors report some interesting metabolic phenotype of the Vps34 Het KI mice, but there are multiple fundamental issues (as described below) and data not compelling enough to support their conclusion.

We are pleased with the overall assessment of our study by this Reviewer who clearly captured the importance of using gene 'knock-in' over gene 'knock-out' approaches for Vps34 studies, especially for modelling pharmacological kinase-inhibition. He/she further raised the following comments:

Major comments:

1. The authors claim that Vps34 Het KI mice show enhanced insulin sensitivity, but (as described below) there is no clear evidence supporting this notion. Please clarify what is the definition of insulin sensitivity in the authors' point of view.

Our definition of increased insulin-sensitivity is an improved insulin-stimulated clearance of blood glucose.

a. Based on the ITT results shown in Fig. 2D, the authors concluded that KI mice are more insulin sensitive compared to control animals. However, if you actually look at the data, KI mice have lower basal glucose, but upon insulin injection, kinetic (time-course) and magnitude of changes in blood glucose levels look nearly identical between control and KI mice. Please show the data as delta (%) changes from the baseline of respective genotype. If the glucose curve is comparable between control and KI mice, do the authors still claim KI mice are insulin sensitive? If so, please justify.

As suggested by the Referee, we have now presented the data (Fig. 2d, 3d) as delta (% of baseline) of the respective genotype. The revised figures (included below) clearly show an

enhanced insulin responsiveness in Vps34 KI mice compared to WT mice, both under normal chow diet (Fig. 2d) and high-fat diet (Fig 3d). We trust that this addresses this key concern.

Updated Fig. 2d:

Updated Fig. 3d:

In Response to Referee 3, we have now also calculated the glucose disappearance rate from the ITT tests (k_{ITT} ; %/min) as follows: $k_{ITT} = (0.693 \times 100) / t_{1/2}$ where $t_{1/2}$ is the "half-life" calculated from the slope of the plasma glucose concentration. We found that the WT and Vps34 KI mice had a k_{ITT} of 1.35 and 2.77%/min, respectively, indicating a faster glucose clearance in the Vps34 mutant mice, consistent with a better insulin sensitivity.

b. Assuming that insulin sensitivity is similar between control and KI (based on delta % change data), I will need to question the authors' interpretation regarding GTT. What is the potential reason that the KI animals are more glucose tolerant (assuming insulin sensitivity is similar between genotypes as argued above)?

As we have now shown, insulin sensitivity is improved in Vps34 KI mice.

Please show that data 1st phase insulin secretion (plasma insulin levels following 5-10 min glucose injection) and/or show muscle/liver p-Akt blots.

The muscle/liver p-Akt blots were presented in the original version of the manuscript (muscle and liver: Supplementary Fig. 6e,f; primary myotubes and hepatocytes: Fig. 9a,b). These data show a similar insulin-mediated Akt signalling in Vps34 KI and WT cells/tissues.

If the authors claim that basal AMPK levels might be increased and this explains higher glucose disposal per se, please show p-AMPK/p-ACC blots.

We have now added p-AMPK/p-ACC blots for muscle and liver (inserted in original blot in Fig. S6e, f - highlighted in the blots below), documenting enhanced basal AMPK signalling in Vps34 KI tissues compared to WT controls, confirming the *in vitro* data on primary cells shown in the original Fig. 9a and b.

Supplementary Fig. S6e,f

It would be also informative if the authors perform AICAR tolerance test and to see if the KI mice show blunted response (blood glucose kinetic).

This is an excellent suggestion, we have now performed an AICAR tolerance test, shown in the figure below. These data demonstrate a clearly blunted response to AICAR in $Vps34^{D761A/+}$ mice compared to WT mice. This suggests that the increased basal AMPK levels in $Vps34^{D761A/+}$ tissues indeed underlie the higher glucose disposal in these mice. We have now included these data as a new panel of Fig. 9g and inserted the following text in the results section (page 6) of the manuscript:

“To assess the involvement of AMPK in the increased insulin-independent increase in glucose disposal in Vps34-deficient muscle cells, blood glucose was assayed after an intraperitoneal injection of the cell-permeable AMPK activator 5-aminoimidazole-4-carboxamide ribonucleoside (AICAR). As shown in Fig. 9g, AICAR caused a significant reduction in blood glucose levels in WT mice, likely due to the combined effects of enhanced muscle glucose uptake activation and inhibition of hepatic glucose production. Interestingly, the hypoglycaemic effect of AICAR was blunted in $Vps34^{D761A/+}$ mice, indicating that the increased basal AMPK levels in $Vps34^{D761A/+}$ tissues are functionally related to the higher glucose disposal in these mice.”

Legend: AICAR tolerance test upon intraperitoneal injection of 0.15 g/kg of AICAR in mice after overnight starvation. $\geq 5-10$ mice/genotype were used. Data represent mean \pm SEM (non-parametric Mann-Whitney t-test) * $p < 0.05$, ** $p < 0.01$, *** $p < 0.001$.

c. Fig. 4: Results of hyperinsulinemic-euglycemic clamp are not interpretable. As the authors do not show validation of “clamp” and also other associated key parameters.

Please show (in Supplementary section): 1) glucose kinetics to demonstrate that glucose levels are clamped at euglycemic levels and there is no difference between the genotypes, 2) show glucose infusion rate and whole body glucose disposal data (Rd) as well.

We now show the requested parameters in a new Supplementary Table 3 (shown below). The key parameters presented in the Table show that the basal and hyperinsulinaemic glucose levels are clamped at euglycemic levels and there is no difference between the genotypes.

Supplementary Table 3. Glucose kinetics and other metabolic parameters validating the hyperinsulinemic-euglycaemic clamp shown in Fig. 4.

		Normal Chow Diet								
		Basal Rd			Insulin Rd			Insulin GIR		
$\mu\text{mol/kg}\cdot\text{min}$		Mean	sem	t-test	Mean	sem	t-test	Mean	sem	t-test
WT		20.90	2.20	0.44	57.01	14.42	0.58	38.65	7.41	0.92
Vps34^{D761A/+}		18.85	1.37		46.12	5.37		39.97	6.12	

		High Fat Diet								
		Basal Rd			Insulin Rd			Insulin GIR		
$\mu\text{mol/kg}\cdot\text{min}$		Mean	sem	t-test	Mean	sem	t-test	Mean	sem	t-test
WT		12.21	0.73	0.19	27.19	2.06	0.94	18.44	4.69	0.30
Vps34^{D761A/+}		15.23	1.54		26.90	2.68		27.10	4.19	

2. The authors show that AMPK phosphorylation is increased in myotubes/hepatocytes from Vps34 Het KI mice/ Vps34 inhibitor-treated cells, and they claim this would be the key mechanism by which Vps34-deficiency/inhibition causes enhanced insulin sensitivity and protection from high fat diet induced embolic abnormalities. However, there are several questionable observations/interpretations that need to be addressed.

a. What is the mechanism by which AMPK is activated via Vps34-deficiency/inhibition?

Our data suggest that Vps34 inactivation alters cellular energy metabolism by modulating the availability of key substrates for mitochondrial respiration, leading to a cellular stress. We propose that altogether, the reduced mitochondrial respiration and reduced ATP levels in Vps34 KI cells lead to AMPK activation. In the previous version of our manuscript, we had included a summary scheme of the overall mechanism as a supplementary figure, we have now moved this scheme to the main Figure 10d. This should help the Reader in capturing the data from the manuscript.

The authors show there is no change in ATP content in muscle/liver of Vps34 KI mice. Would this be different if authors measure ATP (AMP) levels in myotubes/hepatocytes (due to reduced mitochondria respiratory capacity) from Vps34 KI mice? If so, how do the authors reconcile in vivo and in vitro data?

Fig. 6e in the original manuscript documented a ~25% reduction in ATP levels in Vps34 KI liver compared to WT, with no significant changes in ATP levels in the muscle. Possible explanations for this were mentioned in the text - we stated that this could be due to compensatory

mechanisms in the muscle and/or the notion that muscle, unlike the liver, can produce ATP from sources other than mitochondria (glycolysis and possibly phosphocreatinine) to cope with its high energy demands, possibly masking the reduction in mitochondrial ATP production.

To circumvent compensatory mechanisms induced by long-term inactivation of Vps34, we next tested whether *acute* Vps34 inhibition alters ATP levels in the murine C2C12 myoblast and the Hepa1.6 hepatoma cell lines. As shown in the figure below, inhibition of Vps34 dramatically reduced the ATP levels in these cells. This was also observed by Seahorse analysis in oligomycin-treated myotubes (original Fig. 6b,c). This indicates that Vps34 inhibition can reduce ATP levels in cells of muscle origin. We have now inserted these new data and relevant text in the revised manuscript (Supplementary Fig. 5e and text page 4) as follows: *'To circumvent compensatory mechanisms that could be induced by long-term inactivation of Vps34, we next tested whether acute pharmacological inhibition of Vps34 altered ATP levels in the murine C2C12 myoblast and Hepa1.6 hepatoma cell lines. As shown in Supplementary Fig. 5e, inhibition of Vps34 dramatically reduced the ATP levels in these cells, further indicating that Vps34 activity can control ATP levels in cells of muscle origin.'*

New Supplementary Fig. 5e

Legend: Decreased ATP levels in Hepa.16 hepatoma and C2C12 myoblast cells upon Vps34 inactivation. Cells were cultured in presence or absence of 1 µM Vps34-IN1 overnight in Complete media (CM) or starvation media. ATP levels were determined using ATP bioluminescence kit (Roche).

b. The authors suggest that enhanced glucose uptake in Vps34-deficient muscle cells is mediated via AMPK-TBC1D1 axis and show increased p-S660 TBC1D1 signal. There is no evidence that this site has any functional significance. Other site S237/1 has been better characterized with some genetic evidence associated (O'Neill et al, PNAS, 2011, Frosig, J Physiol, 2010, Chen et al, Diabetologia, 2017).

We have now tested phosphorylation of the TBC1D1^{S237} site as requested (see figure below - with quantification), showing increased phosphorylation of pTBC1D1^{S237}, in line with expectations. We have inserted this figure as a new Supplementary Fig. S6e and modified the text as follows (**change highlighted**): *'In agreement with this, a significant enhancement in the phosphorylation of AMPK substrates, such as Acetyl-CoA carboxylase (ACC; on S79) and TBC1D1 (on S660 and S237) was observed in the basal state...'*

modified Supplementary Fig. S6e

c. Fig 9c and d: Vps34-deficient muscle cells have higher basal AMPK activity and insulin-independent 2DG uptake. They also show that Vps34 inhibitor causes an increase in 2DG in the absence of insulin. Based on these data with enhanced AMPK signaling coupled, they conclude that enhanced insulin-independent glucose uptake in muscle is likely AMPK-dependent via Vps34 inhibition. If that is the case, Vps34 inhibitor-mediated increase in 2DG should be abolished in Vps34-deficient muscle cells. It might be the case that Vps34 is simply a mitochondrial poison which indirectly activates AMPK independently of Vps34. Moreover, to verify that insulin-independent increase in glucose uptake in Vps34-deficient muscle cells is due to AMPK (Fig. 9c), please include AICAR/A769662 and show there is no or blunted increase by these activators (alternatively AMPK α 1/ α 2 can be knocked down by RNAi if feasible in the authors lab).

We agree with the Reviewer that the proposed experiment would be an elegant way to demonstrate that insulin-independent increase in glucose uptake in Vps34-deficient muscle cells is due to AMPK. We agree that Vps34 inhibitor-mediated increase in 2DG should be abolished in Vps34-deficient muscle cells. However, this would only be the case upon homozygous Vps34 kinase-inactivation but not upon heterozygous inactivation as in our experiments, in which the remaining WT Vps34 allele will still respond to Vps34 inhibitor. We would therefore expect a Vps34 inhibitor to still have an impact on 2DG uptake in our cells.

However, we believe that the new data of the AICAR tolerance test (see comment #1b and new Fig. 9e) which reveal a blunted AICAR response, provide valid alternative support for the involvement of higher basal AMPK activity in glucose uptake.

Reviewer #2 (Remarks to the Author):

The study evaluates a whole body heterozygous loss-of-function Vps34 mutant regarding a number of metabolic parameters. Mutants presented increased glucose tolerance and reduced predisposition for diet-induced hepatic steatosis. The phenotype was mostly due to reduced hepatic autophagy, which limited substrate availability of mitochondrial respiration and gluconeogenesis. Some of the metabolic outcomes were reproduced with a pharmacological inhibitor of Vsp34. In general this is a fine work providing important advance in the field and identifying a potentially interesting target for the treatment of diabetes. Because of the potential impact of these findings, some additional experiments and controls are required.

Major comments

1. One major concern about the outcomes of partially inhibiting Vsp34 is that mitochondria function is considerably modified. This could have a short term beneficial effect; however, in the long run this could severely impact on metabolic fitness. It is very important that ageing mutants are analyzed for: adiposity, blood lipids, fat deposition in blood vessels and muscle and markers of

oxidative stress. Also, longevity should be determined (a simple Kaplan-Meier graph would be fine).

We agree that this type of study would be very interesting but believe - in line with the Editor's opinion – that this experiment is out of scope for the present study and within the given timeframe.

2. As repeatedly recommended in the series of guidelines for methods in autophagy research, transmission electron microscopy (TEM) is the gold-standard method for evaluation of autophagosome (Klionski et al Autophagy 2016 12:1-222). Critical experiments evaluating the impact of loss-of-function mutation of Vps34 should include TEM.

This is a fair comment, but we would add that TEM evaluation can be subjective, and thus requires expert analysis, and quantitative approaches for subtle phenotypes.

As part of our study, we had in fact performed multiple experiments using serial block-face scanning electron microscopy (or SBF SEM) on liver tissues to generate high resolution 3-dimensional images as well as transmission electron microscopy (TEM) on liver and various muscle types and primary hepatocytes, in both starved and fed conditions.

However, we did not detect distinctive differences (including morphologically abnormal mitochondria and/or abnormal intracellular compartments) from qualitative analysis of images between the Vps34 heterozygotes and WT samples. We have therefore decided not to include any EM data in our manuscript.

For the perusal of the Referee and Editor, we here provide the results of our quantitative in-depth analysis of primary hepatocytes in fed and starved (EBSS 1h) conditions. We trust that the very labour-intensive nature of these experiments will be appreciated - this analysis has taken us several months to perform, using the most advanced EM techniques.

Given that the results obtained were very subtle, and in our view would not add to the manuscript, we would therefore prefer not to include these data in the revised manuscript, unless advised otherwise by the Referee/Editor. However, we have now mentioned our observations in a statement in the results section of the manuscript (page 4) as follows: *'Qualitative and quantitative electron microscopy analysis did not reveal any robust quantitative differences between WT and Vps34^{D761A/+} primary hepatocytes under starved conditions, confirming our data (Fig. 5e-g; Supplementary Fig. 4b-d) that 50% inactivation of Vps34 kinase activity does not fully abolish the formation of autophagosomes and autophagolysosomes under starvation (data not shown)'*.

Our results are summarised below:

We took images of all autophagosome (AP) and AP-like structures in 5 cells for each condition, sampling from 3 separate experiments on WT and Vps34 KI hepatocytes. The cells were chosen by placing a 20x20 grid over a low magnification TEM image of a single section, and using a random number generator to select individual grid squares. The entirety of a cell intersecting the chosen square was imaged.

From our existing data, we reasoned that there might be subtle changes in the number of forming (open) autophagosomes and therefore we developed an approach to quantify open autophagosomes and the length of the autophagosome membrane. We considered that APs containing material inside similar in granular appearance to the cytoplasm and lacking electron-dense material as "open" AP-like structures. In contrast, the APs containing electron-dense material were considered as "closed" (Fig. a below). Qualitative and

quantitative EM analysis did not reveal any robust quantitative differences between WT and Vps34^{D761A/+} primary hepatocytes under starved conditions, confirming our data that 50% inactivation of Vps34 does not abolish the formation of autophagosomes and autophagolysosomes under starvation (Fig. a and b below). We also found that the length of the AP membrane structures in Vps34^{D761A/+} primary hepatocytes was comparable to that in control cells. However, we found a significant increase in the number of autophagosomes which were not closed. Our quantitative analysis revealed a ~5-fold increase of open AP-like structures that do not contain mitochondria in Vps34^{D761A/+} primary hepatocytes compared to WT under basal condition (Fed, complete medium) (Fig. b). In EBSS there is an increased tendency towards a similar phenotype which is not significantly different. This correlates with our data reporting a decrease of WIPI2 staining in Vps34^{D761A/+} primary hepatocytes compared to WT (Fig. 5e). We speculate that the mild reduction in PI3P (Fig. 5d) seen upon heterozygous inactivation of Vps34 may affect the closure of the AP and LC3 lipidation and recruitment.

Legend: EM analysis of WT and Vps34^{D761A/+} primary hepatocytes. (a) Primary murine hepatocytes were cultured in complete medium (CM), fixed, and subjected to conventional EM analysis. Representative images are shown. Arrows depict the open AP-like structures and arrowheads show the closed AP-like structures. (b) Quantification of the EM images of WT and Vps34^{D761A/+} primary hepatocytes cultured in complete medium (CM) or in starvation medium (EBSS) for 1 h.

Materials & Methods: Quantification of autophagosome-like structures in cultured hepatocytes

Primary hepatocytes were fixed in 2.5% glutaraldehyde/4% paraformaldehyde in 0.1 M PB for 60 min. After fixation, the coverslips were washed several times in 0.1 M PB and post-fixed in 1.5% potassium

ferricyanide/1% osmium tetroxide for 1 h, prior to further washing in PB and incubation in 1% tannic acid in 0.05 M PB for 45 min to enhance membrane contrast. After a brief incubation in 1% sodium sulphate in 0.05 M PB, the coverslips were washed twice in distilled water, and dehydrated through an ascending series of Ethanol (30%, 50%, 2x70%, 2x90%, 2x100%), and propylene oxide before infiltration with Epoxy resin and polymerisation overnight at 60°C. The coverslips were removed from the resin blocks by plunging briefly into liquid nitrogen.

The polymerised blocks were trimmed by hand using a single edged razor blade to form a trapezoid block face for serial ultrathin sectioning. Using a diamond knife, serial ultrathin sections of approximately 70 nm thickness were collected on 1% formvar coated single slot grids. The sections were counterstained with lead citrate to further enhance contrast prior to viewing in the electron microscope (FEI Tecnai G2 Spirit BioTWIN with Gatan Orius CCD camera).

For quantification of 'open' and 'closed' autophagosome-like structures, cells were chosen at random by placing a 20x20 grid over a low magnification image of a single section, and using a random number generator to select individual grid squares. The entirety of a cell intersecting the chosen square was imaged. The imaging conditions and depth of section (with respect to distance from the coverslip surface) were kept consistent between samples. All autophagosome-like structures in each of 5 cells from WT and Vps34^{D761A/+} cultures (n=3 for each) were examined. The autophagosome-like structures were classified as 'open' when the contents of the structure were similar in appearance to the surrounding cellular cytoplasm; 'closed' structures displayed a distinct difference in the density and/or granularity of their contents when compared to the surrounding cytoplasm. The grid overlay for cell selection and viewing was done using ImageJ software.

3. The kinase activity shown in figure 1b should include a positive control, another PI3K isoform.

It is not clear to us which of the eight PI3K isoforms the Reviewer would suggest we test. Importantly, the fact that we did not observe differences in Akt signalling between WT and Vps34 KI cells strongly suggests that the kinase activity of at least the class I PI3K isoforms is not altered.

4. Why in figure 2b fasting glucose is about 80 mg/dl for both WT and mutant, whereas in the other experiments (Fig 2a and 2d) glucose is different between WT and mutant?

We believe this is due to inherent variation in mouse cohorts used. Please note that the fasting blood glucose levels in Figures 2a and 2c are similar.

5. Authors should calculate kITT for the experiment shown in figure 2d. for stating that mutants are more sensitive to insulin glucose decay during ITT is more important than actual glucose levels.

This comment relates to point#1 raised by Reviewer#1. As explained above, we have now presented the data as delta (% of baseline) of the respective genotype. The revised Fig. 2d and Fig.3d clearly show an enhanced insulin responsiveness in Vps34 KI mice compared to WT mice.

As requested by this Reviewer, we have now also calculated the glucose disappearance rate from the ITT tests (kITT; %/min) as follows: $k_{ITT} = (0.693 \times 100) / t_{1/2}$ where $t_{1/2}$ is the "half-life" calculated from the slope of the plasma glucose concentration. We found that the WT and Vps34 KI mice had a k_{ITT} of 1.35 and 2.77%/min, respectively, indicating a faster glucose clearance in the Vps34 mutant mice, consistent with a better insulin sensitivity.

Minor

1. In abstract (line 51) and discussion (line 370) authors propose that drugs that inhibit Vps34 could be potentially useful in patients that are intolerant to metformin. They say that about 10% of patients are intolerant to metformin but no reference is presented. In fact, most patients (in my experience much more than 90%) tolerate metformin and the only real major problem associated with the use of this drug is lactic acidosis, which is an extremely rare event. Please consider reviewing this information.

Gastro-intestinal intolerance to metformin is relatively common and is a major reason for non-compliance and drug withdrawal, especially with the normal release formulation. A 5-10% level of intolerance is widely cited, including in much of the product literature, and is in keeping with the authors' clinical experience. To reflect the FDA summary of product characteristics accurately, however, the 10% figure is now replaced by 5% (see Garber *et al.*, *Am.J.Med.* 1997; and <https://www.fda.gov/Drugs/DrugSafety/ucm493244.htm>; and Bristol-Myers Squibb Company, NDA 20-357/S-031, NDA 21-202/S-016).

We would also emphasize the point that most current guidelines recommend withdrawing metformin when the glomerular filtration rate falls below 30 ml/minute. This precludes the use of metformin in a large number of patients with diabetes and renal impairment, whether related to diabetic nephropathy, vascular disease, or other underlying renal disease. A new drug that was not renally cleared would thus have large potential value in these patients.

Based on this feedback, we therefore updated the text in the discussion section as follows: 'Around 5% of diabetic patients do not tolerate Metformin, due mostly to gastro-intestinal side effects, such as nausea and/or diarrhoea, while in many more patients where Metformin is effective it is withdrawn in the face of declining renal or cardiac function due to fears of lactic acidosis (see Ref.41 and <https://www.fda.gov/Drugs/DrugSafety/ucm493244.htm>). Although it is not expected that Vps34 inhibitors would replace Metformin, we expect this class of inhibitors to be complementary.'

2. It seems there is a splice in the bottom immunoblots for LC3 in Supplementary Figure 4. If that is the case, the splice should be clearly labeled or the experiment repeated in order to obtain a satisfactory image in a single gel.

The splice spotted by the Reviewer was an artefact generated during the scanning process of the exposed film of the western blot. We have now rescanned the film and updated the figure accordingly in the revised manuscript.

Reviewer #3 (Remarks to the Author):

In this study Bilanges and coworkers addresses the impact of Vps34 (Vps34 phosphoinositide 3-kinase (PI3K)), a central player of autophagy, on energy homeostasis. The authors report that systemic Vsp34 inactivation in mice leads to enhanced insulin sensitivity and improved glucose tolerance, with reduced hepatic glucose production and increased glucose uptake in muscle. At the mechanistic level, Vps34 inactivation dampened hepatic autophagy, limiting substrate availability for mitochondrial respiration and gluconeogenesis. In muscle, Vps34 inactivation triggers a metabolic switch from oxidative phosphorylation towards glycolysis and enhanced glucose uptake. Interestingly, Vps34 inactivation did not affect Akt signaling but activated the AMPK pathway,

suggesting potential clinical utility for Vps34 inhibitors in Type-2 diabetic patients. This a very well performed study with potential therapeutic outcomes.

Major points

1. In Figure 2, the ITT between the two genotypes seems rather similar. How would it look expressed in percent of base line? The area under the curve should be measured.

This point is similar to comment 1 of Reviewer#1, the updated Figures 2d and 3d clearly show that Vps34 KI mice are more insulin sensitive than WT mice.

2. Figure 3. It is surprising that insulin concentrations were not increased in response to HFD (i.e fasting hyperinsulinemia ?).

We do not understand this comment, given that the level of plasma insulin is significantly increased (>35 fold) in response to HFD (please compare fasting insulinaemia under NCD~0.4ng/ml (Figure 2b) to fasting insulinaemia under HFD~14ng/ml (Figure 3b).

In the same line, it is surprising not to have any increase in liver weight upon HFD treatment.

We also were surprised to observe only a mild increase in liver mass in response to HFD, this is possibly related to % of HFD (45%) used in our study as compared to the 60% which is most often used in metabolic studies.

3. The observation that circulating levels of adiponectin is interesting. What is the

This comment is not clear to us, we believe the sentence above was unfinished.

4. Glucose uptake is increased in skeletal muscle, with a only a tendency in white and brown adipose tissue. To exclude the participation of adipose tissues to the phenotype, insulin signaling experiments (such as Akt, IRS1/2 phosphorylation assays etc..) should be performed in white and brown adipose tissues, and compared to muscle and liver.

We had performed signalling studies on white adipose tissue upon insulin stimulation and did not find differences between WT and Vps34 KI mice. These data, which were not included in the original manuscript, are in line with our clamp studies (Fig. S3i).

We have now included these signalling data (new Supplementary Fig. S6g) and added the highlighted text as follows: “Both basal and insulin-stimulated activation of Akt [as assessed by phosphorylation of Akt on S473 and T308) and its downstream targets such as GSK3 α / β (on S21 and S29), AS160 (on S642) and PRAS40 (on T246)] and mTORC1 [assessed by phosphorylation of S6K (on T389) and S6 (on S240 and 244)], were unaffected in Vps34D761A/+ myotubes (Fig. 9a; Supplementary Fig. 6a) and hepatocytes (Supplementary Fig. 6b-d), as well as in muscle, liver and white adipose tissue (WAT) in vivo (Supplementary Fig. 6e-g). Similar observations were made upon treating WT hepatocytes with the Vps34-IN1 inhibitor (Supplementary Fig. 6b)”.

Unfortunately, we do not have data available for the BAT.

New Supplementary Fig. S6g

5. Key genes of the b-oxidation pathway should be measured (PPAR α , LCPT1 etc..). Does increased in b-oxidation rates explain the lack of steatosis in response to HFD diet?

To address this question, we have now analysed expression of genes involved in β -oxidation (PPAR α , Cpt1b, Cpt1c and PGC1 α) by qPCR (TaqMan).

Pharmacological inhibition of Vps34 in the Hepa1.6 hepatoma cell line led to a 2- to 4.5-fold increase in expression of these genes (Figure below). Genetic inhibition of Vps34 in liver also led to substantial upregulation of these genes (Figure below). These data support the notion that Vps34 inactivation increased fatty acid oxidation in hepatocytes.

These data are now included as new figures 9e,f in the revised manuscript with updated text in the result section (pages 5-6) as follows:

"ACC is a rate-controlling enzyme in the synthesis of malonyl-CoA, a critical precursor for fatty acid biosynthesis and a potent inhibitor of mitochondrial fatty acid β -oxidation. Malonyl-CoA is an intermediate in fatty acid synthesis and an allosteric inhibitor of carnitine palmitoyltransferase 1, which regulates the transfer of long-chain acyl-CoAs from the cytosol into the mitochondria. AMPK-mediated phosphorylation of ACC on S79 inhibits its activity and therefore the upregulation of pACCS79 levels observed in Vps34^{D761A/+} cells might result in decreased lipogenesis and/or increased fatty acid oxidation. Gene expression levels of key players in β -oxidation showed that Vps34 inactivation in cells and tissues increased gene expression of the transcriptional activators PPAR α and PGC-1 α and their target genes including Cpt1b and Cpt1c (Fig. 9e,f), suggesting that Vps34 inactivation increases fatty acid oxidation in liver."

And in the discussion section (page 7) as follows: *"Interestingly, Vps34^{D761A/+} hepatocytes also showed a ~20-25% increase in mitochondria content and an increased expression of genes encoding mitochondrial proteins. This may in part be due to an increased PGC-1 α -mediated response given that PGC-1 α plays a central role in mitochondria biogenesis. Such an increased AMPK-PGC-1 α -mediated response in Vps34^{D761A/+} hepatocytes may explain the ~20-25% increase in mitochondria content and increased expression of genes encoding mitochondrial proteins in these cells."*

New Figures 9e,f

Legend: Expression of genes involved in fatty acid β -oxidation. Hepa1.6 cells were cultured overnight in presence of 1 μ M Vps34-IN1 in starvation medium. Liver samples from WT and Vps34^{D761A/+} were collected. RNA was extracted from lysed cells or tissues and subjected to reverse transcription reaction prior to qPCR using TaqMan technology.

Reviewers' comments:

Reviewer #1 (Remarks to the Author):

The authors addressed majority of the key issues that I had raised and the manuscript has been improved after the revision. I have one remaining minor concern regarding the interpretation of AICAR tolerance test data: "...As shown in Fig. 9g, AICAR caused a significant reduction in blood glucose levels in WT mice, likely due to the combined effects of enhanced muscle glucose uptake activation and inhibition of hepatic glucose production. Interestingly, the hypoglycaemic effect of AICAR was blunted in Vps34D761A/+ mice, indicating that the increased basal AMPK levels in Vps34D761A/+ tissues are functionally related to the higher glucose disposal in these mice". AICAR-induced hypoglycemic effect is similar between the genotypes, but recover/over-shooting (blood glucose) was significantly faster/higher in the D761A/+ mice. Given that the Vps34 deficiency occurs at global levels, it might be the case that glucagon response (e.g. in liver) was more robust in the KI group. I would suggest the authors be modest regarding the interpretation of the data and include potential alternative interpretation since they do not have measurement of insulin, glucagon and muscle/liver AMPK signalling in this experiment.

Reviewer #2 (Remarks to the Author):

Virtually none of the suggestions were attended.

Reviewer #3 (Remarks to the Author):

No more comments

NCOMMS-16-30432 - "Vps34 PI 3-kinase inactivation enhances insulin sensitivity through reprogramming of mitochondrial metabolism"

Reviewer #1 (Remarks to the Author):

'The authors addressed majority of the key issues that I had raised and the manuscript has been improved after the revision'

We are pleased with this response of Reviewer#1.

Minor comments:

'I have one remaining minor concern regarding the interpretation of AICAR tolerance test data: "...As shown in Fig. 9g, AICAR caused a significant reduction in blood glucose levels in WT mice, likely due to the combined effects of enhanced muscle glucose uptake activation and inhibition of hepatic glucose production. Interestingly, the hypoglycaemic effect of AICAR was blunted in Vps34D761A/+ mice, indicating that the increased basal AMPK levels in Vps34D761A/+ tissues are functionally related to the higher glucose disposal in these mice". AICAR-induced hypoglycemic effect is similar between the genotypes, but recover/over-shooting (blood glucose) was significantly faster/higher in the D761A/+ mice. Given that the Vps34 deficiency occurs at global levels, it might be the case that glucagon response (e.g. in liver) was more robust in the KI group. I would suggest the authors be modest regarding the interpretation of the data and include potential alternative interpretation since they do not have measurement of insulin, glucagon and muscle/liver AMPK signalling in this experiment'

This is a fair comment and we have therefore modified the text along the Referee comment, as follows: "Interestingly, the AICAR-induced hypoglycaemic effect was similar between genotypes, but the blood glucose levels recovered significantly faster in Vps34^{D761A/+} mice. This could be explained by a higher glucose disposal and/or a possible better hepatic glucagon response in Vps34^{D761A/+} mice".

Reviewer #2 (Remarks to the Author):

'Virtually none of the suggestions were attended'.

We strongly disagree with this Reviewer as we feel it is an unfair and incorrect comment, given that – amongst other - we presented data of ~6-months electron microscopy experiments in response to one of his/her comments.

However, in light of our EM observations (requested by this Referee) and in line with Editorial feedback based on this finding, we agree that we have to be much more prudent in our claims on the involvement of autophagy. We have therefore rewritten the abstract, and made substantial modifications to the introduction and the discussion (highlighted in the revised text).

Reviewer #3 (Remarks to the Author):

'No more comments'